# Mixtures Closest to a Given Measure: A Semidefinite Programming Approach

## Abstract

Mixture models, such as Gaussian mixture models (GMMs), are widely used in machine learning to represent complex data distributions. A key challenge, especially in high-dimensional settings, is to determine the mixture order and estimate the mixture parameters. We study the problem of approximating a target measure, available only through finitely many of its moments, by a mixture of distributions from a parametric family (e.g., Gaussian, exponential, Poisson), with approximation quality measured by the 2-Wasserstein ($W_2$) or the total variation (TV) distance. Unlike many existing approaches, the parameter set is not assumed to be finite; it is modeled as a compact basic semi-algebraic set. We introduce a hierarchy of semidefinite relaxations with asymptotic convergence to the desired optimal value. In addition, when a certain rank condition is satisfied, the convergence is even finite and recovery of an optimal mixing measure is obtained. We also present an application to clustering, where our framework serves either as a stand-alone method or as a preprocessing step that yields both the number of clusters and strong initial parameter estimates, thereby accelerating convergence of standard (local) clustering algorithms.

## 1 Introduction

Modeling data using mixtures of probability distributions has become a well-established approach across statistics, data science, and machine learning. Mixture models have been successfully employed in a wide range of tasks, including classification (Permuter et al., 2006; Zio et al., 2007), image and signal processing (Reynolds et al., 2000; Stauffer & Grimson, 1999; Yu et al., 2012), natural language processing (Athiwaratkun & Wilson, 2017; Clinchant & Perronnin, 2013), and clustering (Chen & Zhang, 2024; Fraley & Raftery, 2002; Karlis & Meligkotsidou, 2007), among others. For a comprehensive overview of (finite) mixture models and their theoretical foundations, we refer the reader to (Antoniak, 1974; McLachlan & Peel, 2000; Titterington et al., 1985).

Mixture models, and in particular Gaussian mixtures, have been widely investigated in both theory and practice, yet important aspects of their behavior remain incompletely understood. One of the most persistent challenges is determining the number of components, or mixture order. Because mixtures with different orders can often produce nearly indistinguishable empirical distributions, the task is typically to recover the smallest number of components that remains consistent with the observed data.

**On the number of components:** Several approaches have been proposed for estimating the number of components in a mixture model (McLachlan & Rathnayake, 2014). A common strategy involves placing a prior (e.g. Gaussian) on the form of the mixture and selecting the model using likelihood-based criteria. For example, hypothesis testing via likelihood ratio tests (Ghosh & Sen, 1985; Lo, 2005) provides a powerful framework for model selection. However, these tests are highly sensitive to violations of standard regularity conditions (Drton, 2009; Seidel et al., 2000), which frequently occur in mixture models due to issues such as non-identifiability and boundary effects.

Beyond hypothesis testing, other likelihood-based criteria aim to minimize a form of Kullback-Leibler divergence between the true data-generating distribution and the fitted model. Among these, AIC (Akaike, 1974) and BIC (Schwarz, 1978) are widely used. However, both rely heavily on the quality of the maximum likelihood estimate, which is often affected by local optima in

the non-convex likelihood landscape of mixture models. As a result, the selected models may either overfit or underfit the data. The bootstrap-based criterion EIC (Ishiguro et al., 1997) offers a potential improvement by addressing overfitting, but it comes at the cost of increased computational complexity and instability in high-dimensional settings.

Nonparametric methods for selecting the number of components, such as silhouette statistic (Kaufman & Rousseeuw, 1990) or gap statistic (Tibshirani et al., 2001), are widely used because they are model-free and relatively simple to compute. However, their heuristic nature and reliance on distance measures make them prone to instability and poor performance, especially in noisy or high-dimensional data. Techniques from numerical algebraic geometry (Shirinkam et al., 2020) have also been applied.

**On parameter estimation:** Likelihood-based methods, particularly *maximum likelihood estimation* (MLE) combined with the Expectation-Maximization (EM) algorithm (Dempster et al., 1977), are widely used for parameter estimation in mixture models. Over the years, numerous enhancements have made this approach the standard tool for tackling such problems. However, computing the global MLE remains challenging due to the non-convexity of the likelihood surface. Consequently, the EM algorithm often converges only to local optima and is quite sensitive to heuristic choices for initialization and stopping criteria. Its slow convergence and need to access the full dataset at each iteration also make it computationally expensive in large or high-dimensional settings (Karlis & Xekalaki, 2003; Redner & Walker, 1984). The Nonparametric Maximum Likelihood Estimator (NPMLE) (Kiefer & Wolfowitz, 1956) extends classical MLE by optimizing over all probability measures, allowing flexible estimation of mixing distributions without fixing the number of components. While it admits a discrete solution in one dimension (Lindsay, 1995), its structure in higher dimensions is less understood and computationally challenging due to the infinite-dimensional nature of the problem. Recent advances aim to improve scalability and adaptivity without relying on fixed grids (Yan et al., 2023).

Alternatively, mixture parameters can be estimated using the *method of moments*, which involves matching or minimizing the discrepancy between empirical and population moments (Hall, 2005; Hansen, 1982). Non-convexity of moment equations coupled with identifiability issues (Améndola et al., 2018) are the main drawbacks of this approach. A number of recent improvements, some of which rely on techniques from tensor decompositions, on both theoretical and algorithmic perspective can be found in (Ge et al., 2015; Heinrich & Kahn, 2018; Khouja et al., 2022; Lindberg et al., 2025; Pereira et al., 2022; Wu & Yang, 2020).

While it is impossible to provide an exhaustive list of techniques for mixture parameter estimation, we highlight several recent contributions, primarily from the theoretical computer science community, that develop polynomial-time recovery algorithms under various separation conditions (Arbas et al., 2023; Bakshi et al., 2022; Hosseini & Sra, 2015; Kothari et al., 2022; Liu & Li, 2022; Moitra & Valiant, 2010).

Most prior work emphasizes worst-case complexity and specifies algorithms at a high level with limited empirical evaluation; additionally, the mixture order is often treated as known and fixed *a priori*, or else determined by external criteria rather than learned within the model.

**Contribution:** We study the problem of approximating a target probability measure, available only through finitely many of its moments and not necessarily compactly supported, by a mixture of probabilities from a parametric family, which minimizes either the 2-Wasserstein distance ($W_2$) or the total variation (TV) distance. In contrast to much prior work, the admissible parameter space is not assumed finite; instead, it is modeled as a compact basic semi-algebraic set. Building on and extending recent developments from (Lasserre, 2024a;b), our contribution is twofold:

1. We develop Moment-SOS semidefinite relaxations (Lasserre, 2009) for approximating a given probability measure $\mu$ on $\mathbb{R}^n$ by a mixture $\nu$ that minimizes either the $W_2$ or the TV distance. In our formulation, the mixing measure is the decision variable, and we add a polynomial *regularization term* on its moments to improve numerical conditioning and eventually promote sparsity. As a result, we can recover finite mixtures *without* fixing the mixture order or imposing strong prior assumptions on the mixture structure. Our approach rests on two observations:

   - For the parametric families we consider (e.g., Gaussian, Poisson, exponential), their moments are *polynomial functions* of the parameters, and we prove (Lemma A.1) that the resulting mixtures are *moment-determinate*. This structure appears to be overlooked in

the optimization and machine-learning literature, whereas we show that it leads to polynomial constraints on the unknown mixture that are directly amenable to SDP lifting, with (sometimes finite) *global convergence* guarantees.

- Computing the TV is notoriously challenging. We leverage a domination property of the Hahn-Jordan decomposition of the signed measure $\mu - \nu$ to derive tractable relaxations and establish convergence. To our knowledge, this device has not been explicitly exploited in theoretical and computational formulations of TV for mixture models.

2. In a first potential application, and for illustration purposes, we *design an algorithm* that inherits strong convergence properties and can be applied to a range of (unsupervised) clustering tasks under varying assumptions on the underlying parametric mixture family. Our method can significantly improve the performance of standard clustering algorithms, such as $k$-means (Arthur & Vassilvitskii, 2007) and EM (Dempster et al., 1977), by reducing both their number of iterations and their sensitivity to initialization.

## 1.1 NOTATION

Let $\mathbb{R}[\boldsymbol{x}] = \mathbb{R}[x_1, \ldots, x_n]$ denote the ring of real polynomials in $n$ variables, where $\boldsymbol{x} = (x_1, \ldots, x_n)$. For any degree $d \in \mathbb{N}$, let $\mathbb{R}_d[\boldsymbol{x}]$ represent the vector space of real polynomials of degree at most $d$, equipped with the standard monomial basis $\boldsymbol{v}_d := (1, x_1, \ldots, x_n, x_1^2, \ldots, x_n^d)$ of size $s(n,d) := \binom{n+d}{d}$. A positive semidefinite (PSD) matrix $\mathbf{M}$ is denoted by $\mathbf{M} \succeq 0$. The space of symmetric matrices of size $n$ is denoted $\mathcal{S}^n$, and its PSD cone by $\mathcal{S}^n_+$. Identity matrix of size $n$ is denoted by $\mathbf{I}_n$.

For a given pair of integers $(d,n) \in \mathbb{N}^2$, we define the multi-index set $\mathbb{N}^n_d := \{\boldsymbol{\alpha} = (\alpha_1, \ldots, \alpha_n) \in \mathbb{N}^n \mid \sum_{i=1}^n \alpha_i \leq d\}$. Accordingly, for any $\boldsymbol{x} \in \mathbb{R}^n$ and $f \in \mathbb{R}_d[\boldsymbol{x}]$, we write $f(\boldsymbol{x}) = \sum_{\boldsymbol{\alpha} \in \mathbb{N}^n_d} f_{\boldsymbol{\alpha}} \boldsymbol{x}^{\boldsymbol{\alpha}}$, where each coefficient $f_{\boldsymbol{\alpha}} \in \mathbb{R}$, and $\boldsymbol{x}^{\boldsymbol{\alpha}} = x_1^{\alpha_1} \cdots x_n^{\alpha_n}$. The support of $f$ is defined as $\operatorname{supp}(f) := \{\boldsymbol{\alpha} \in \mathbb{N}^n \mid f_{\boldsymbol{\alpha}} \neq 0\}$. We denote by $\mathbb{R}^S$ the space of real-valued sequences indexed by a set $S$. The space of finite positive Borel measures (resp. probabilities) supported on a set $K$ is denoted by $\mathcal{M}_+(K)$ (resp. $\mathscr{P}(K)$).

## 1.2 MOMENT AND LOCALIZING MATRICES

Let $\boldsymbol{y} = (y_{\boldsymbol{\alpha}})_{\boldsymbol{\alpha} \in \mathbb{N}^n}$ be a real-valued sequence. We can always associate to $\boldsymbol{y}$ the linear *Riesz functional* $L_{\boldsymbol{y}}$ defined as follows:

$$L_{\boldsymbol{y}} : \mathbb{R}[\boldsymbol{x}] \ni f \mapsto L_{\boldsymbol{y}}(f) := \sum_{\boldsymbol{\alpha} \in \operatorname{supp}(f)} f_{\boldsymbol{\alpha}} y_{\boldsymbol{\alpha}} \in \mathbb{R}. \tag{1}$$

For every $d \in \mathbb{N}$, the symmetric matrix $\mathbf{M}_d(\boldsymbol{y})$ of size $s(n,d)$, with rows and columns indexed by $\mathbb{N}^n_d$, and defined by

$$\mathbf{M}_d(\boldsymbol{y})(\boldsymbol{\alpha}, \boldsymbol{\beta}) := L_{\boldsymbol{y}}(\boldsymbol{x}^{\boldsymbol{\alpha}} \boldsymbol{x}^{\boldsymbol{\beta}}) = y_{\boldsymbol{\alpha}+\boldsymbol{\beta}}, \quad \boldsymbol{\alpha}, \boldsymbol{\beta} \in \mathbb{N}^n_d, \tag{2}$$

is called the *pseudo-moment matrix* of order $d$ associated with $\boldsymbol{y}$. Similarly, for $r \in \mathbb{R}[\boldsymbol{x}]$, the symmetric matrix $\mathbf{M}_d(r\boldsymbol{y})$ of size $s(n,d)$, with rows and columns indexed by $\mathbb{N}^n_d$, and defined by

$$\mathbf{M}_d(r\boldsymbol{y})(\boldsymbol{\alpha}, \boldsymbol{\beta}) := L_{\boldsymbol{y}}(r(\boldsymbol{x}) \boldsymbol{x}^{\boldsymbol{\alpha}} \boldsymbol{x}^{\boldsymbol{\beta}}) = \sum_{\boldsymbol{\gamma} \in \operatorname{supp}(r)} r_{\boldsymbol{\gamma}} y_{\boldsymbol{\alpha}+\boldsymbol{\beta}+\boldsymbol{\gamma}}, \quad \boldsymbol{\alpha}, \boldsymbol{\beta} \in \mathbb{N}^n_{d-\lceil \deg(r)/2 \rceil}, \tag{3}$$

is called the *localizing matrix* of order $d$ associated with $r$ and $\boldsymbol{y}$.

A sequence $\boldsymbol{y}$ has a *representing measure* $\mu$ if $y_{\boldsymbol{\alpha}} = \int_{\mathbb{R}^n} \boldsymbol{x}^{\boldsymbol{\alpha}} \, d\mu(\boldsymbol{x})$, for all $\boldsymbol{\alpha} \in \mathbb{N}^n$. In this case, necessarily $\mathbf{M}_d(\boldsymbol{y}) \succeq 0$ for all $d \in \mathbb{N}$. Furthermore, if the support of $\mu$ is contained in the semi-algebraic set $\{\boldsymbol{x} \in \mathbb{R}^n \mid r_j(\boldsymbol{x}) \geq 0, \ j \in \{1, \ldots, l\}\}$, then $\mathbf{M}_d(r_j \boldsymbol{y}) \succeq 0$ for all $d \in \mathbb{N}$ and all $j \in \{1, \ldots, l\}$.

## 2 PROBLEM FORMULATION

Let $\mu$ be an (unknown) probability measure on $\mathbb{R}^n$, $n \in \mathbb{N}^*$. In many settings with heterogeneous data, it is natural to approximate $\mu$ by a mixture of distributions from a parametric family

$\{\mu_{\boldsymbol{\theta}}\}_{\boldsymbol{\theta} \in S_{\boldsymbol{\theta}}} \subset \mathscr{P}(\mathbb{R}^n)$. We make a non-restrictive assumption that parameter vector $\boldsymbol{\theta}$ lies in a basic semi-algebraic set

$$S_{\boldsymbol{\theta}} := \left\{ \boldsymbol{\theta} \in \mathbb{R}^p \mid r_j(\boldsymbol{\theta}) \geq 0, \ j \in \{1, \ldots, l\} \right\}, \ r_j \in \mathbb{R}_d[\boldsymbol{\theta}], \ p, l, d \in \mathbb{N}^*. \tag{4}$$

As a concrete example, consider a mixture of multivariate Gaussian distributions where the parameters of each Gaussian component lie, for example, in the set

$$S_{\boldsymbol{m}, \boldsymbol{\Sigma}} := \left\{ (\boldsymbol{m}, \boldsymbol{\Sigma}) \in \mathbb{R}^n \times \mathcal{S}_+^n \mid a\mathbf{I}_n \preceq \boldsymbol{\Sigma} \preceq b\mathbf{I}_n, \ r_j(\boldsymbol{m}) \geq 0, \ j \in \{1, \ldots, l\} \right\}, \tag{5}$$

for some $0 < a < b$ and polynomials $r_j \in \mathbb{R}[\boldsymbol{m}]$. The semi-algebraic structure of $S_{\boldsymbol{m}, \boldsymbol{\Sigma}}$ arises from the fact that the PSD constraints on the covariance matrix $\boldsymbol{\Sigma}$ can be expressed using at most $2n$ polynomial inequalities of degree at most $n$, by requiring the principal minors of matrices $\boldsymbol{\Sigma} - a\mathbf{I}_n$ and $b\mathbf{I}_n - \boldsymbol{\Sigma}$ to be nonnegative.

Let $\phi \in \mathscr{P}(S_{\boldsymbol{\theta}})$ be a *mixing measure*, and define the induced mixture $\nu_\phi \in \mathscr{P}(\mathbb{R}^n)$ by

$$\nu_\phi(B) \ = \ \int_{S_{\boldsymbol{\theta}}} \mu_{\boldsymbol{\theta}}(B) \, d\phi(\boldsymbol{\theta}) \quad \text{for all } B \in \mathcal{B}(\mathbb{R}^n). \tag{6}$$

Our goal is to approximate $\mu$ by some $\nu_\phi$ from this mixture family, according to a chosen probability distance $\mathrm{dist}(\cdot, \cdot)$ (e.g., $W_2$ or TV), while adding a small regularization on the mixing measure. Therefore, we consider the following optimization problem

$$\tau_{\varepsilon, R}^{\mathrm{dist}} := \inf_{\nu_\phi \in \mathscr{P}(\mathbb{R}^n), \, \phi \in \mathscr{P}(S_{\boldsymbol{\theta}})} \left\{ \mathrm{dist}(\mu, \nu_\phi) + \varepsilon \int_{S_{\boldsymbol{\theta}}} R(\boldsymbol{\theta}) \, d\phi(\boldsymbol{\theta}) \right\}, \tag{7}$$

where $R \in \mathbb{R}[\boldsymbol{\theta}]$ is a user-chosen *polynomial* regularizer and $\varepsilon > 0$ controls the regularization strength.

Importantly, we do *not* assume that $\phi$ is $K$-atomic. However, finite mixtures that are optimal may instead arise from the optimization procedure itself.

In practice, $\mu$ is observed via $N$ samples $\boldsymbol{x}_1, \ldots, \boldsymbol{x}_N \in \mathbb{R}^n$, giving the empirical measure $\mu^N = \frac{1}{N} \sum_{i=1}^N \delta_{\boldsymbol{x}_i}$, with $\delta_{\boldsymbol{x}_i}$ the Dirac mass at $\boldsymbol{x}_i$. However, even if $\mu$ is replaced by $\mu^N$, Problem (7) remains infinite-dimensional, as it is not restricted to finite-order mixtures, and since we do not assume that $S_{\boldsymbol{\theta}}$ is finite. It is therefore intractable in its native form. To obtain tractable relaxations, we work with sequences of finite size representing *moments* of $\mu$ and of the mixing measure $\phi$.

Recall that a moment sequence is called *determinate* if it admits a unique representing measure. A classical sufficient condition is the multivariate Carleman condition (Nussbaum, 1965). We assume that the data-generating measure $\mu$, with moment sequence $\boldsymbol{\mu} = (\mu_{\boldsymbol{\alpha}})_{\boldsymbol{\alpha} \in \mathbb{N}^n}$, is moment-determinate:

**Assumption 2.1** (Moment determinacy). *For all $j \in \{1, \ldots, n\}$,*

$$\sum_{k=1}^{+\infty} \left( L_{\boldsymbol{\mu}}(x_j^{2k}) \right)^{-1/2k} = +\infty.$$

We also require parametric distributions $\{\mu_{\boldsymbol{\theta}}\}_{\boldsymbol{\theta} \in S_{\boldsymbol{\theta}}}$ to have moments that depend polynomially on parameters, a property satisfied by many standard parametric families (e.g., Gaussian, Poisson, exponential mixtures):

**Assumption 2.2** (Polynomial moments). *For every $\boldsymbol{\alpha} \in \mathbb{N}^n$, there exists a polynomial $p_{\boldsymbol{\alpha}} \in \mathbb{R}_{|\boldsymbol{\alpha}|}[\boldsymbol{\theta}]$ such that*

$$\int_{\mathbb{R}^n} \boldsymbol{x}^{\boldsymbol{\alpha}} \, d\nu_\phi(\boldsymbol{x}) = \int_{S_{\boldsymbol{\theta}}} p_{\boldsymbol{\alpha}}(\boldsymbol{\theta}) \, d\phi(\boldsymbol{\theta}) \quad \text{for all } \phi \in \mathscr{P}(S_{\boldsymbol{\theta}}).$$

For example, univariate distribution $\mu_{m, \sigma} \sim \mathcal{N}(m, \sigma^2)$ is such that $p_3(m, \sigma) = m^3 + 3m\sigma^2$. Under Assumptions 2.1 and 2.2, problem (7) can be transformed into a *generalized moment problem* (GMP) with polynomial data. Hence, $\tau_{\varepsilon, R}^{\mathrm{dist}}$ can be approximated arbitrarily well by a Lasserre-type semidefinite programming hierarchy (Lasserre, 2009; 2015).

More precisely, at relaxation order $d \in \mathbb{N}^*$, we solve an SDP that only involves moments of $\mu$ up to order $2d$, yielding a lower bound $\tau_{d, \varepsilon, R}^{\mathrm{dist}} \leq \tau_{\varepsilon, R}^{\mathrm{dist}}$. As $d$ increases, the sequence $(\tau_{d, \varepsilon, R}^{\mathrm{dist}})_d$ is monotone non-decreasing and converges to $\tau_{\varepsilon, R}^{\mathrm{dist}}$ under mild conditions, with *finite* convergence whenever a standard rank (*flat extension*) condition holds. In polynomial optimization, such finite convergence is generic (Lasserre, 2015, Chapter 7).

In the finite-convergence regime, we further show how to extract the mixture order and parameter estimates from the optimal SDP solutions, which makes the SDP hierarchy a practical preprocessing tool for mixture estimation and, in particular, for clustering.

## 3 SDP RELAXATIONS: MAIN PROPERTIES

In this section, we explain how the regularized optimization problem (7) gives rise to tractable semidefinite relaxations; the overall pipeline is summarized in Figure 1. The construction is the same for all distances considered: only the distance-specific linear and PSD constraints change. At a high level, starting from a target measure $\mu$ (or its empirical counterpart $\mu^N$) and a parametric mixture family $\{\mu_{\boldsymbol{\theta}}\}_{\boldsymbol{\theta} \in S_{\boldsymbol{\theta}}}$, we:

1. Write the chosen distance $\mathrm{dist} \in \{\mathrm{W}_2, \mathrm{TV}\}$ between $\mu$ and a mixture $\nu_\phi$ as an infinite-dimensional linear program over measures;

2. Use the polynomial moment representation from Assumption 2.2 to reformulate the problem as a generalized moment problem (GMP), in which the decision variables are the *infinite* moment sequences of the unknown measures;

3. Consider only the first $2d$ moments and treat the resulting truncated pseudo-moment sequences as new decision variables. Then impose the *necessary* conditions for these pseudo-moments to admit representing measures supported on the prescribed sets by requiring the associated moment and localizing matrices to be positive semidefinite. This yields a finite-dimensional SDP of type (8) whose size grows with $d$.

All relaxations we solve share the same generic SDP structure (explicit derivations are deferred to Appendices A.3 and A.4). Let $\boldsymbol{y}_d^{\mathrm{dist}} \in \mathbb{R}^{m_{n,p,d}}$, with $m_{n,p,d} \in \mathbb{N}$, collect all truncated moments (up to degree $2d$) of the decision measures. The corresponding SDP relaxation of (7) has the form:

$$\tau_{d,\varepsilon,R}^{\mathrm{dist}} = \min_{\boldsymbol{y}_d^{\mathrm{dist}}} \langle \mathbf{c}_{d,\varepsilon,R}^{\mathrm{dist}}, \boldsymbol{y}_d^{\mathrm{dist}} \rangle \quad \text{such that} \quad \mathbf{A}_d^{\mathrm{dist}} \boldsymbol{y}_d^{\mathrm{dist}} = \mathbf{b}_d^{\mathrm{dist}}, \ \mathbf{M}_d^{\mathrm{dist}}(\boldsymbol{y}_d^{\mathrm{dist}}) \succeq 0. \tag{8}$$

Observe that (8) is indeed an SDP instance: it consists of minimizing the linear functional $\langle \mathbf{c}_{d,\varepsilon,R}^{\mathrm{dist}}, \boldsymbol{y}_d^{\mathrm{dist}} \rangle$ over an affine subspace $\{\boldsymbol{y}_d^{\mathrm{dist}} : \mathbf{A}_d^{\mathrm{dist}} \boldsymbol{y}_d^{\mathrm{dist}} = \mathbf{b}_d^{\mathrm{dist}}\}$ intersected with the inverse image of the positive semidefinite cone $\mathcal{S}_+^{m_{n,p,d}}$ under the affine map $\boldsymbol{y}_d^{\mathrm{dist}} \mapsto \mathbf{M}_d^{\mathrm{dist}}(\boldsymbol{y}_d^{\mathrm{dist}})$, where $\mathbf{M}_d^{\mathrm{dist}}(\boldsymbol{y}_d^{\mathrm{dist}})$ is a symmetric block-diagonal matrix whose blocks are distance-specific moment and localizing matrices (Lasserre, 2015).

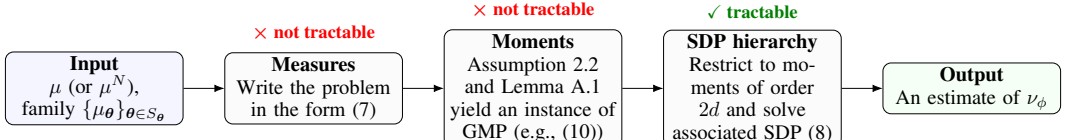

Figure 1: Mixture estimation via SDPs - Pipeline overview.

### 3.1 WASSERSTEIN DISTANCE FOR MIXTURES

We first instantiate our framework with the *squared* 2-Wasserstein distance. For *given* $\mu, \nu_\phi \in \mathscr{P}(\mathbb{R}^n)$, this distance is defined in the following way:

$$\mathrm{W}_2(\mu, \nu_\phi)^2 := \inf_{\lambda \in \mathscr{P}(\mathbb{R}^n \times \mathbb{R}^n)} \left\{ \int_{\mathbb{R}^n \times \mathbb{R}^n} \|\boldsymbol{x} - \boldsymbol{y}\|^2 \, d\lambda(\boldsymbol{x}, \boldsymbol{y}) \mid \pi_{\boldsymbol{x}} \# \lambda = \mu, \ \pi_{\boldsymbol{y}} \# \lambda = \nu_\phi \right\}, \tag{9}$$

where $\pi_{\boldsymbol{x}}, \pi_{\boldsymbol{y}} : \mathbb{R}^n \times \mathbb{R}^n \to \mathbb{R}^n$ are the canonical projections and $\pi_{\boldsymbol{x}} \# \lambda, \pi_{\boldsymbol{y}} \# \lambda$ denote the induced push-forward measures.

Since we aim at approximating $\mu$ by a mixture, we optimize over all mixing measures $\phi$ supported on $S_{\boldsymbol{\theta}}$. Inspired by Lasserre (2024a), we introduce the following $\varepsilon$-*regularized moment formulation*:

$$\tau_{\varepsilon,R}^{\mathrm{W}_2} := \begin{cases} \inf_{\lambda \in \mathscr{P}(\mathbb{R}^n \times \mathbb{R}^n), \ \phi \in \mathscr{P}(S_{\boldsymbol{\theta}})} L_{\boldsymbol{\lambda}}(\|\boldsymbol{x} - \boldsymbol{y}\|^2) + \varepsilon L_{\boldsymbol{\phi}}(R) & \text{(10a)} \\ \text{s.t.} \quad \lambda_{\boldsymbol{\alpha},\mathbf{0}} = \mu_{\boldsymbol{\alpha}}, \quad \lambda_{\mathbf{0},\boldsymbol{\alpha}} = L_{\boldsymbol{\phi}}(p_{\boldsymbol{\alpha}}), \ \boldsymbol{\alpha} \in \mathbb{N}^n, & \text{(10b)} \end{cases}$$

where, for all $\boldsymbol{\alpha}, \boldsymbol{\beta} \in \mathbb{N}^n$, $\lambda_{\boldsymbol{\alpha},\boldsymbol{\beta}} := \int_{\mathbb{R}^n \times \mathbb{R}^n} \boldsymbol{x}^{\boldsymbol{\alpha}} \boldsymbol{y}^{\boldsymbol{\beta}} \, d\lambda(\boldsymbol{x}, \boldsymbol{y})$ and $p_{\boldsymbol{\alpha}}$ is given by Assumption 2.2.

Problem (10) is still an infinite-dimensional linear program. To obtain a tractable relaxation, we truncate the moment sequences at degree $2d$ and replace the unknown measures by finite collections

of pseudo-moments. We then impose the usual positivity and support conditions via moment and localizing matrices (Lasserre, 2009; 2015), which are necessary for these (pseudo) moments to admit representing measures supported on the prescribed sets.

Let $d_j := \lceil \deg(r_j)/2 \rceil$ for $j \in \{0, \dots, l\}$ and set $d_{\min} := \max\{ \max_{j \in \{1,\dots,l\}} d_j, \lceil \deg(R)/2 \rceil \}$. For any $d \geq d_{\min}$, we introduce the following decision vector:

$$\boldsymbol{y}_d^{\mathrm{W}_2} := (\boldsymbol{\lambda}, \boldsymbol{\phi}) \in \mathbb{R}^{m_{n,p,d}}, \ m_{n,p,d} = 2s(n, 2d) + s(p, 2d), \tag{11}$$

where $\boldsymbol{\lambda}$ and $\boldsymbol{\phi}$ are indexed by all monomials in $(\boldsymbol{x}, \boldsymbol{y})$ and $\boldsymbol{\theta}$, respectively, up to total degree $2d$, and play the role of truncated moment sequences of the measure $\lambda$ and the mixing measure $\phi$.

We then define the $\varepsilon$-*regularized* $\mathrm{W}_2$-*moment relaxation of order* $d$, namely:

$$\tau_{d,\varepsilon,R}^{\mathrm{W}_2} := \begin{cases} \min\limits_{\boldsymbol{y}_d^{\mathrm{W}_2}} \ \langle \mathbf{c}_{d,\varepsilon,R}^{\mathrm{W}_2}, \boldsymbol{y}_d^{\mathrm{W}_2} \rangle & \text{(12a)} \\[2mm] \text{s.t.} \qquad \mathbf{A}_d^{\mathrm{W}_2} \boldsymbol{y}_d^{\mathrm{W}_2} = \mathbf{b}_d^{\mathrm{W}_2}, \quad \mathbf{M}_d^{\mathrm{W}_2}(\boldsymbol{y}_d^{\mathrm{W}_2}) \succeq 0. & \text{(12b)} \end{cases}$$

In (12), the cost vector $\mathbf{c}_{d,\varepsilon,R}^{\mathrm{W}_2}$ collects the coefficients of the objective $L_{\boldsymbol{\lambda}}(\|\boldsymbol{x} - \boldsymbol{y}\|^2) + \varepsilon L_{\boldsymbol{\phi}}(R)$; the linear operator $\mathbf{A}_d^{\mathrm{W}_2}$ enforces the marginal moment-matching constraints $\lambda_{\boldsymbol{\alpha},\mathbf{0}} = \mu_{\boldsymbol{\alpha}}$ and $\lambda_{\mathbf{0},\boldsymbol{\alpha}} = L_{\boldsymbol{\phi}}(p_{\boldsymbol{\alpha}})$ for $|\boldsymbol{\alpha}| \leq 2d$; and $\mathbf{M}_d^{\mathrm{W}_2}(\boldsymbol{y}_d^{\mathrm{W}_2})$ is a symmetric block-diagonal matrix whose blocks are the moment matrix $\mathbf{M}_d(\boldsymbol{\lambda})$ and the localizing matrices $\mathbf{M}_{d-d_j}(r_j \boldsymbol{\phi})$, $j \in \{0, \dots, l\}$.

From every feasible solution of (10), one can construct a finite-dimensional vector that is feasible for (12). Hence, (12) is a genuine convex relaxation and $\tau_{d,\varepsilon,R}^{\mathrm{W}_2} \leq \tau_{\varepsilon,R}^{\mathrm{W}_2}$ for all $d \geq d_{\min}$.

**Remark 3.1.** *The regularization polynomial $R$ both stabilizes the SDP numerically and encodes structural preferences on the mixture. A typical choice is $R(\boldsymbol{\theta}) = \sum_{0 \neq \boldsymbol{\gamma} \in \mathbb{N}_d^p} \boldsymbol{\theta}^{2\boldsymbol{\gamma}}$, which helps promote a low-rank structure on the optimal solution, thereby reducing model complexity and avoiding overfitting. Finally, in practice, only empirical moments $\mu_{\boldsymbol{\alpha}}^N$ are available, so one may allow a small error margin for the moment-matching constraints. By the law of large numbers, the discrepancy $\mu_{\boldsymbol{\alpha}}^N - \mu_{\boldsymbol{\alpha}}$ becomes small as $N$ grows. In high-dimensional settings, however, accurately estimating higher-order moments may still require large sample size. A detailed non-asymptotic analysis relating the admissible relaxation order $d$ to $N$ and $n$ via concentration bounds for empirical moments is provided in (Vu et al., 2022).*

**Theorem 3.2.** *Assume that $S_{\boldsymbol{\theta}}$ in (4) is compact and that Assumptions 2.1 and 2.2 hold. Let $\boldsymbol{y}_d^{\mathrm{W}_2} = (\boldsymbol{\lambda}^{*(d)}, \boldsymbol{\phi}^{*(d)})$ be an optimal solution of (12) for some $d \geq d_{\min}$, and suppose that*

$$\operatorname{rank} \mathbf{M}_d(\boldsymbol{\phi}^{*(d)}) = \operatorname{rank} \mathbf{M}_{d-d_{\min}}(\boldsymbol{\phi}^{*(d)}) = K \tag{13}$$

*for some $K \in \mathbb{N}^*$. Then the hierarchy has converged at level $d$, i.e., $\tau_{d,\varepsilon,R}^{\mathrm{W}_2} = \tau_{\varepsilon,R}^{\mathrm{W}_2}$. In addition, one can recover a $K$-atomic mixing measure $\phi^*$ supported on $S_{\boldsymbol{\theta}}$ such that the associated mixture $\nu_{\phi^*}$ minimizes the $\varepsilon$-regularized $\mathrm{W}_2$ distance to $\mu$.*

Condition (13), usually referred to as the *flat extension* criterion, thus provides a certificate of finite convergence for the hierarchy. In practice, Theorem 3.2 (proved in Appendix A.3) implies that the mixture order $K$ can be read off from the rank of $\mathbf{M}_d(\boldsymbol{\phi}^{*(d)})$. Moreover, the corresponding component parameters can be recovered by a linear-algebraic extraction procedure (Curto & Fialkow, 2005; Henrion & Lasserre, 2005) (see Appendix A.2, Algorithm 2).

### 3.2 Total variation distance for mixtures

Alternatively, we may measure the proximity between the unknown data-generating measure and a mixture of distributions using the total variation distance. For *given* $\mu, \nu_\phi \in \mathscr{P}(\mathbb{R}^n)$ with all moments finite and satisfying Assumption 2.1, the total variation distance is defined as follows:

$$\tau^{\mathrm{TV}} := \|\mu - \nu_\phi\|_{\mathrm{TV}} = 2 \sup_{B \in \mathcal{B}(\mathbb{R}^n)} |\mu(B) - \nu_\phi(B)|. \tag{14}$$

In Lemma A.1, which can be considered as a contribution of independent interest, we prove that commonly used mixture families satisfy Assumption 2.1. Moreover, using the Hahn-Jordan decomposition (see Appendix A.2) of the signed measure $\mu - \nu_\phi$, one can show that $\tau^{\mathrm{TV}}$ is also the optimal value of the infinite-dimensional linear program below:

$$\tau^{\mathrm{TV}} = \inf_{\psi_+, \psi_- \in \mathscr{M}_+(\mathbb{R}^n)} \left\{ \psi_+(1) + \psi_-(1) \mid \psi_+ - \psi_- = \mu - \nu_\phi, \ \psi_+ \leq \mu, \ \psi_- \leq \nu_\phi \right\}. \tag{15}$$

Optimizing over all mixing measures $\phi \in \mathscr{P}(S_{\boldsymbol{\theta}})$ then leads to the following $\varepsilon$-*regularized formulation* for computing the TV-optimal mixtures:

$$\tau_{\varepsilon,R}^{\mathrm{TV}} := \begin{cases} \inf\limits_{\psi_+,\psi_- \in \mathscr{M}_+(\mathbb{R}^n),\ \phi \in \mathscr{P}(S_{\boldsymbol{\theta}})} \psi_+(1) + \psi_-(1) + \varepsilon L_{\boldsymbol{\phi}}(R) & \text{(16a)} \\ \text{s.t.} \qquad \psi_+ - \psi_- = \mu - \nu_{\phi}, \quad \psi_+ \leq \mu, \quad \psi_- \leq \nu_{\phi}. & \text{(16b)} \end{cases}$$

Domination constraints $\mu \leq \nu_{\phi}$ (i.e., $\nu_{\phi} - \mu \in \mathcal{M}_+(\mathbb{R}^n)$) translate to the moment matrices: $\mu \leq \nu_{\phi} \iff \mathbf{M}_d(p;\boldsymbol{\phi}) \succeq \mathbf{M}_d(\boldsymbol{\mu})$, for all $d \in \mathbb{N}$, where $\mathbf{M}_d(p;\boldsymbol{\phi})(\boldsymbol{\alpha},\boldsymbol{\beta}) := L_{\boldsymbol{\phi}}(p_{\boldsymbol{\alpha}+\boldsymbol{\beta}})$, $\boldsymbol{\alpha},\boldsymbol{\beta} \in \mathbb{N}_d^n$, $d \in \mathbb{N}$, stands for the moment matrix of the unknown mixture $\nu_{\phi}$.

Therefore, for any $d \geq d_{\min}$ we collect all truncated moments (up to degree $2d$) of $\psi_+$, $\psi_-$ and $\phi$ into a single vector $\boldsymbol{y}_d^{\mathrm{TV}} := (\boldsymbol{\psi}_+, \boldsymbol{\psi}_-, \boldsymbol{\phi}) \in \mathbb{R}^{m_{n,p,d}}$, and we define the $\varepsilon$-*regularized* TV-*moment relaxation of order* $d$:

$$\tau_{d,\varepsilon,R}^{\mathrm{TV}} := \begin{cases} \min\limits_{\boldsymbol{y}_d^{\mathrm{TV}}} \langle \mathbf{c}_{d,\varepsilon,R}^{\mathrm{TV}}, \boldsymbol{y}_d^{\mathrm{TV}} \rangle & \text{(17a)} \\ \text{s.t.} \qquad \mathbf{A}_d^{\mathrm{TV}} \boldsymbol{y}_d^{\mathrm{TV}} = \mathbf{b}_d^{\mathrm{TV}}, \quad \mathbf{M}_d^{\mathrm{TV}}(\boldsymbol{y}_d^{\mathrm{TV}}) \succeq 0. & \text{(17b)} \end{cases}$$

SDP problem (17) is therefore determined by:

- the cost vector $\mathbf{c}_d^{\mathrm{TV}}$ representing the objective $\psi_+(1) + \psi_-(1) + \varepsilon L_{\boldsymbol{\phi}}(R)$;

- the linear operator $\mathbf{A}_d^{\mathrm{TV}}$ encoding, for every multi-index $\boldsymbol{\alpha} \in \mathbb{N}_{2d}^n$, the moment-matching equality $\psi_{+,\boldsymbol{\alpha}} - \psi_{-,\boldsymbol{\alpha}} - \mu_{\boldsymbol{\alpha}} + L_{\boldsymbol{\phi}}(p_{\boldsymbol{\alpha}}) = 0$:

- the block-diagonal matrix $\mathbf{M}_d^{\mathrm{TV}}(\boldsymbol{y}_d^{\mathrm{dist}})$, with blocks enforcing $\mathbf{M}_d(\boldsymbol{\mu}) \succeq \mathbf{M}_d(\boldsymbol{\psi}_+) \succeq 0$, $\mathbf{M}_d(p;\boldsymbol{\phi}) \succeq \mathbf{M}_d(\boldsymbol{\psi}_-) \succeq 0$, and $\mathbf{M}_{d-d_j}(r_j\boldsymbol{\phi}) \succeq 0$ for $j \in \{0,\dots,l\}$.

**Theorem 3.3.** *Assume that $S_{\boldsymbol{\theta}}$ in (4) is compact and that Assumptions 2.1 and 2.2 hold. Let $\boldsymbol{y}_d^* = (\boldsymbol{\psi}_+^{*(d)}, \boldsymbol{\psi}_-^{*(d)}, \boldsymbol{\phi}^{*(d)})$ be an optimal solution of (17) for some $d \geq d_{\min}$, and assume that*

$$\operatorname{rank} \mathbf{M}_d(\boldsymbol{\phi}^{*(d)}) = \operatorname{rank} \mathbf{M}_{d-d_{\min}}(\boldsymbol{\phi}^{*(d)}) = K \tag{18}$$

*for some $K \in \mathbb{N}^*$. Then the hierarchy stabilizes at level $d$, i.e., $\tau_{d,\varepsilon,R}^{\mathrm{TV}} = \tau_{\varepsilon,R}^{\mathrm{TV}}$, and there exists a $K$-atomic mixing measure $\phi^*$ supported on $S_{\boldsymbol{\theta}}$ such that the induced mixture $\nu_{\phi^*}$ solves the $\varepsilon$-regularized TV-distance approximation problem (16) with target $\mu$.*

If the sufficient rank-condition in Theorem 3.3 is not satisfied, one can still show that $\lim_{d\to\infty} \tau_{d,\varepsilon,R}^{TV} = \tau_{\varepsilon,R}^{TV}$, where the measure domination constraints (16b) are crucial for proving convergence (Appendix A.4, Theorem A.2). In addition, every accumulation point $\phi^*$ (not necessarily atomic) of the sequence $(\phi^{*(d)})_{d\geq d_{\min}}$ defines a mixture $\nu_{\phi^*}$ that is optimal for the $\varepsilon$-regularized formulation defined in (16).

## 3.3 FROM SDP TO CLUSTERING: ALGORITHM CONSTRUCTION

In light of Theorems 3.2 and 3.3, finite convergence at some relaxation order $d$ can be detected through a rank condition on the optimal moment matrix $\mathbf{M}_d(\boldsymbol{\phi}^{*(d)})$. In practice, however, numerical sensitivities make the exact rank computation unreliable. We therefore estimate the mixture order $\widehat{K} \in \mathbb{N}^*$ using a cumulative-energy criterion applied to the spectrum of $\mathbf{M}_d(\boldsymbol{\phi}^{*(d)})$:

$$\widehat{K} := \min \left\{ r \in \{1,\dots,s(p,d)\} \ \Big| \ \sum_{j=1}^{r} \eta_j \ \geq \ (1-\texttt{tol}) \sum_{j=1}^{s(p,d)} \eta_j \right\}. \tag{19}$$

where $(\eta_j)_{j\in\{1,\dots,s(p,d)\}}$ are eigenvalues of $\mathbf{M}_d(\boldsymbol{\phi}^{*(d)})$ arranged in the decreasing order, and $\texttt{tol} > 0$ is a tolerance threshold. Algorithm 2 allows us then to recover a parameter estimate $\widehat{\boldsymbol{\theta}}_{\mathrm{TV},i}$ or $\widehat{\boldsymbol{\theta}}_{\mathrm{W}_2,i}$ for each $i \in \left\{1,\dots,\widehat{K}\right\}$. We summarize this procedure in Algorithm 1.

---

**Algorithm 1** Mixture order and parameter extraction from SDP relaxations

---

**Require:** Order $d \geq d_{\min}$, moments $\{\mu_{\boldsymbol{\alpha}}^N\}_{|\boldsymbol{\alpha}| \leq 2d}$, theoretical moments $p_{\boldsymbol{\alpha}}$ of a family $\{\mu_{\boldsymbol{\theta}}\}_{\boldsymbol{\theta}}$, support description $(r_j)_{j \in \{1,\ldots,l\}} \in \mathbb{R}[\boldsymbol{\theta}]$, dist $\in \{W_2, TV\}$, regularization $\varepsilon > 0$, tol $> 0$
  1: Depending on dist, solve (12) or (17) at order $d$ and recover $\phi^{*(d)}$
  2: **if** $\operatorname{rank} \mathbf{M}_d(\phi^{*(d)}) = \operatorname{rank} \mathbf{M}_{d-d_{\min}}(\phi^{*(d)})$ **then**
  3:    Estimate $\widehat{K}$ via (19). Extract $\widehat{\boldsymbol{\theta}}_{TV}$ or $\widehat{\boldsymbol{\theta}}_{W_2}$ using Algorithm 2
  4: **else**
  5:    Increase $d$ and repeat from step 1
  6: **end if**
**Ensure:** $\widehat{K}$, and $\left\{\widehat{\boldsymbol{\theta}}_{TV,i}\right\}_{i=1}^{\widehat{K}}$ or $\left\{\widehat{\boldsymbol{\theta}}_{W_2,i}\right\}_{i=1}^{\widehat{K}}$

---

When the flatness condition is not satisfied and further increasing $d$ becomes computationally prohibitive, an approximate solution can be extracted via a more general Gelfand-Naimark-Segal (GNS) construction that applies to *nearly flat* pseudo-moment matrices (Klep et al., 2018). In some cases, the rank condition need only be checked on a principal submatrix of $\mathbf{M}_d(\phi^{*(d)})$ formed by moments involving only a subset of distribution parameters (e.g., means for Gaussian mixtures), reducing size and improving numerical stability.

## 4 EXPERIMENTAL RESULTS

In this section, and as a first potential application of the methodology, we report some preliminary results when using Algorithm 1 in some clustering problems. Exhaustive details of the numerical setup are provided in Appendix C.

**Illustration:** To have a better understanding of Algorithm 1's behavior, we first consider an elementary example with no clusters, namely when $\mu$ is the univariate uniform distribution on $[0, 1]$, accessible via $N = 2000$ randomly drawn samples. We compare our method with likelihood-based approaches (optimized via the EM algorithm) for fitting GMMs, in which the mixture order $\widehat{K}$ is estimated either using the BIC or cross-validation (CV) criterion.

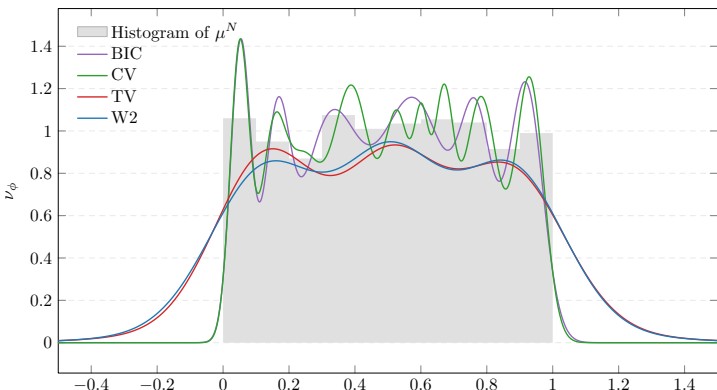

Figure 2: Fitting GMMs to a sample from $d\mu(x) = \mathbb{1}_{[0,1]}(x)\,dx$. Blue and red densities are obtained by computing $\tau_{4,0,R}^{W_2}$ and $\tau_{4,0,R}^{TV}$, respectively, with $S_{m,\sigma} = \left\{(m, \sigma) \mid m \geq m^2, \sigma \geq \sigma^2\right\} = [0, 1]^2$.

From Figure 2, we observe that mixtures recovered by both $W_2$ and TV are smoother (consisting of $\widehat{K} = 4$ components each) than the ones based on EM, even without regularization. Moreover, since we obtained $\tau_{4,0,R}^{W_2} = \tau_{4,0,R}^{TV} = 0$, our relaxations recover mixtures $\nu_{\phi^*}$ whose moments are undistinguishable from those of $\mu^N$, up to order $2d = 8$. Given that parameters estimates recovered by EM-based approaches belong to $S_{m,\sigma}$, we can guarantee that those EM-induced mixtures cannot be better (in $\varepsilon$-regularized $W_2$ or TV-distance sense) than the ones recovered by Algorithm 1.

**Randomly generated instances.** We consider Gaussian mixtures in $\mathbb{R}^2$ with a varying number of components $K$. Two key geometric parameters in this setting are the *separability* $c > 0$, controlling the relative distance between component means, and the *eccentricity* $\epsilon(\mathbf{\Sigma}) > 0$, defined as the ratio of the largest to the smallest eigenvalue of the covariance matrices (Dasgupta, 1999). For $K \in \{2, 5\}$, we generate 50 independent mixture parameter configurations and draw $N = 1000$ samples from each. We then extract $\widehat{K}$, $\left\{\widehat{\boldsymbol{\theta}}_{\mathrm{TV},i}\right\}_{i=1}^{\widehat{K}}$ and $\left\{\widehat{\boldsymbol{\theta}}_{\mathrm{W}_2,i}\right\}_{i=1}^{\widehat{K}}$ using Algorithm 1 with $d = 4$. We initialize both $k$-means and EM algorithm [1] for fitting GMMs with our extracted Gaussian means parameter estimates and assess whether this leads to improved performance compared to random initializations. Mixture parameters were generated to satisfy $c = 5$ (relatively well separated) and $\epsilon(\mathbf{\Sigma}) = 0.25$ (non-spherical distributions). Obtained results are presented in Figure 3.

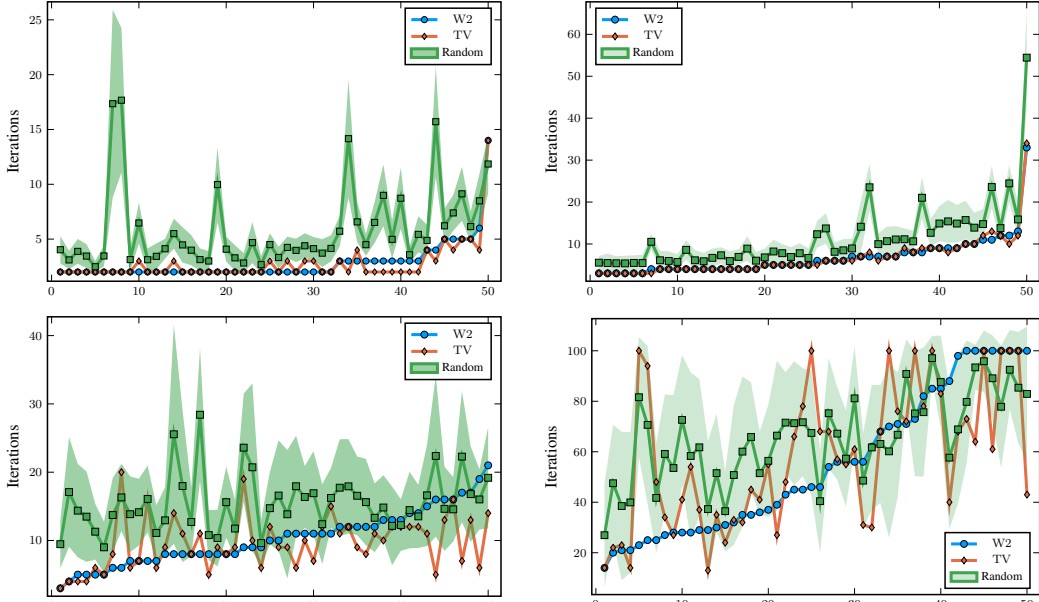

Figure 3: Impact of Algorithm 1-based initialization on the number of iterations required by $k$-means (left column) and EM (right column). Results are shown for mixtures with $K = 2$ (top row) and $K = 5$ (bottom row). Experiments with random initialization were repeated 100 times (green squares (average) and bands (one standard deviation)). The $x$-axis represents mixture indexes, sorted in ascending order of the iteration count under the $\mathrm{W}_2$ initialization. Finally, assembling and solving the $\mathrm{W}_2$-based (resp. TV-based) SDP relaxations takes on average $0.1755\,\mathrm{s}$ (resp. $0.1506\,\mathrm{s}$), whereas the point-extraction step (Algorithm 1) requires only $0.00036\,\mathrm{s}$ on average.

When GMM components are moderately separated and non-spherical, Algorithm 1-based initializations clearly outperform `Random`. For $K = 2$ (top row), both $k$-means and EM converge in very few iterations when initialized with $\widehat{\boldsymbol{m}}_{\mathrm{W}_2}$ or $\widehat{\boldsymbol{m}}_{\mathrm{TV}}$, with little variability across mixtures. In contrast, random initialization is slower and regularly exhibits a very large variance in iteration counts.

Mixtures with $K = 5$ (bottom row) components are more challenging, which is witnessed by increased average iteration counts across all initializations. Our methods, however, remain more efficient around $90\%$ (resp. $67\%$) of time for $k$-means (resp. EM). Indeed, the $\widehat{\boldsymbol{m}}_{\mathrm{W}_2}$-based (resp. $\widehat{\boldsymbol{m}}_{\mathrm{TV}}$-based) $k$-means initializations require on average $31.8\%$ (resp. $36.5\%$) fewer iterations compared to random initialization. Interestingly, EM algorithm appears to favor $\widehat{\boldsymbol{m}}_{\mathrm{W}_2}$-based initializations ($18.63\%$ improvement compared to `Random`) to the ones that are $\widehat{\boldsymbol{m}}_{\mathrm{TV}}$-based ($13.91\%$ improvement compared to `Random`). These findings call for a clearer understanding of which problem classes and algorithmic settings favor one distance type over the other.

---

[1]EM iteratively estimates GMM parameters by alternating between: (*i*) E-step that computes soft assignments of data points to each component and (*ii*) M-step that updates the parameters using these assignments.

**MNIST data.** We evaluate the performance of our approach on subsets of the MNIST dataset ($28 \times 28$ images), applying PCA to retain only 2 principal components. We select Gaussian mixtures as the modeling family in Algorithm 1. Impact on $k$-means performance is given in Figure 4.

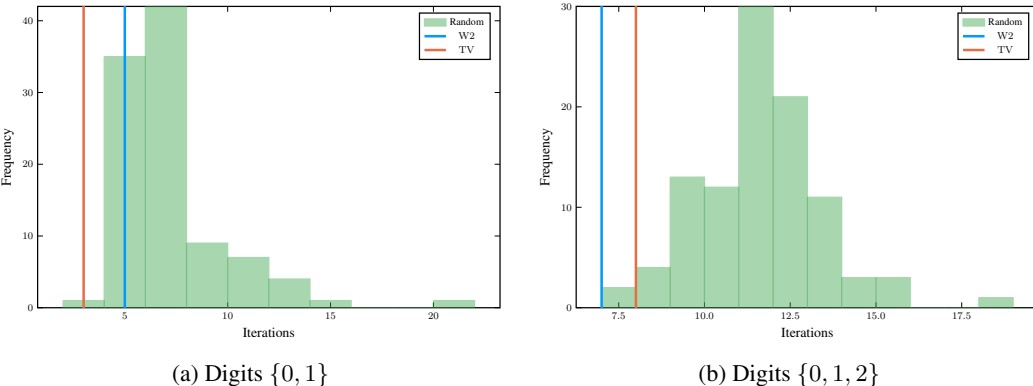

(a) Digits $\{0, 1\}$                                               (b) Digits $\{0, 1, 2\}$

Figure 4: Effect of Algorithm 1-based initialization (with $d = 4$) on the number of iterations required by $k$-means on MNIST, compared against 100 `Random` initializations.

Algorithm 1 initializations reach the same $k$-means objective as random initialization (left: $\approx 4.136 \times 10^2$, right: $\approx 6.734 \times 10^2$) while requiring fewer iterations. For digits $\{0, 1\}$ (resp. $\{0, 1, 2\}$), `Random` needs on average 7.28 (resp. 11.6) iterations. The per-run time of $k$-means scales as $\mathcal{O}(T N K n)$, where $T$ is the number of iterations. Lowering $T$ therefore delivers near-linear savings, which can be very beneficial, especially in high-dimensional settings.

## 5 CONCLUSION AND FUTURE WORKS

In this work, we have introduced two regularized Moment-SOS-type hierarchies of SDP relaxations for approximating (via either $W_2$ or TV distance minimization) an arbitrary given measure by mixtures from specific parametric families. The framework is highly general, requiring only minimal assumptions on the underlying measure. In particular, and in contrast to much of the existing literature, we neither assume that the mixing measure is atomic, nor that the mixture parameters lie in a finite set, nor that the unknown measure is compactly supported. We have established strong convergence guarantees (asymptotic, and sometimes even finite) for the proposed method and illustrated its efficiency through an application to clustering.

One of the main limitations of our approach lies in its moderate scalability: when either $n$ or $p$ is large, the size of the PSD constraints grows rapidly with $d$. Consequently, only low-order relaxations can be solved in practice, which may not provide sufficient precision. A promising direction for future work is to enhance scalability by exploiting distribution-family-specific structure (e.g., via symmetry detection (Riener et al., 2013)) or by reformulating the problem in lower-dimensional spaces, as in (Kolouri et al., 2017), for example. Preliminary results in Appendix B already indicate that this approach can be quite effective.

Moreover, alternative regularization schemes can be explored and their effect on the recovery of optimal solutions systematically assessed. This perspective may help better understand the interplay between $W_2$ and TV distance, and can potentially extend to other probability distances that admit semi-algebraic representations. Quantifying convergence rate with respect to the sample size is another interesting direction.

Finally, given its modeling power, we believe that our approach can be useful beyond clustering, especially in applications where a principled quantitative estimate of the discrepancy between probability measures is of paramount importance.

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

# A    THEORETICAL GUARANTEES

In this section, we explicitly derive the dual formulations of the relaxation problems presented in Section 3. These dual problems are important computationally, since the software we use returns optimal primal solutions from optimal dual variables. We then establish the convergence of the proposed hierarchies.

Before presenting our main results, we introduce several auxiliary results that are essential for a complete understanding of our approach.

## A.1    DUAL SIDE OF THE MOMENT HIERARCHY

To each moment relaxation of the Lasserre hierarchy Lasserre (2009) we can associate a dual sum-of-squares (SOS) relaxation. For sake of completeness, we briefly describe basic ingredients that are necessary to define these dual SOS relaxations.

Recall that a polynomial $f \in \mathbb{R}_{2d}[\boldsymbol{\theta}]$ is called a sum-of-squares (SOS) if there exist $k \in \mathbb{N}^*$ and polynomials $p_1, \ldots, p_k \in \mathbb{R}_d[\boldsymbol{\theta}]$ such that $f = p_1^2 + \cdots + p_k^2$. Furthermore, any SOS polynomial admits a representation of the form $f = \boldsymbol{v}_d^\top \mathbf{G} \boldsymbol{v}_d$, where $\mathbf{G} \in \mathcal{S}_+^{s(p,d)}$ is known as the *Gram matrix*. Let $\Sigma[\boldsymbol{\theta}] \subset \mathbb{R}[\boldsymbol{\theta}]$ (resp. $\Sigma_d[\boldsymbol{\theta}] \subset \mathbb{R}_{2d}[\boldsymbol{\theta}]$) denote the set of SOS polynomials (resp. SOS polynomials of degree at most $2d$). Of course, every SOS polynomial is non-negative. However, the converse is not true.

Recall that $r_0$ denotes the constant polynomial $\mathbb{R}^p \ni \boldsymbol{\theta} \mapsto r_0(\boldsymbol{\theta}) := 1$. To any given family $\{r_1, \ldots, r_l\} \subset \mathbb{R}[\boldsymbol{\theta}]$, we associate the *quadratic module* $\mathcal{Q}(r) = \mathcal{Q}(r_1, \ldots, r_l) \subset \mathbb{R}[\boldsymbol{x}]$ defined by

$$\mathcal{Q}(r) := \left\{ \sum_{j=0}^{l} \sigma_j r_j : \sigma_j \in \Sigma[\boldsymbol{\theta}], \ j \in \{0, \ldots, l\} \right\} \tag{20}$$

and its degree-$2d$ truncated version

$$\mathcal{Q}_d(r) := \left\{ \sum_{j=0}^{l} \sigma_j r_j : \sigma_j \in \Sigma_{d-d_j}[\boldsymbol{\theta}], \ j \in \{0, \ldots, l\} \right\}, \tag{21}$$

where $d_j = \left\lceil \frac{\deg(r_j)}{2} \right\rceil$ for all $j \in \{0, \ldots, l\}$. Obviously, membership in $\mathcal{Q}_d(r)$ is a clear certificate of positivity over the semi-algebraic set $\{\boldsymbol{\theta} \in \mathbb{R}^p \mid r_j(\boldsymbol{\theta}) \geq 0, \ j \in \{1, \ldots, l\}\}$.

## A.2    AUXILIARY RESULTS

**Extracting minimizers.**    Finite convergence of the Lasserre hierarchy occurs whenever the so-called *flat extension* condition is satisfied at some order $d \geq d_{\min}$. In this case, global minimizers of the original problem can be recovered using Algorithm 2 (see (Curto & Fialkow, 2005; Henrion & Lasserre, 2005)), which we reproduce here for completeness.

---

**Algorithm 2** Minimizer extraction

---

**Require:** The moment matrix $\mathbf{M}_d(\boldsymbol{\phi}^{*(d)})$ of rank $K$ satisfying the flatness condition (13).

**Ensure:** Points $\widehat{\boldsymbol{\theta}}_{\text{dist},i} \in S_{\boldsymbol{\theta}}, \ i \in \{1, \ldots, K\}$, that are global global minimizers of (12) or (17).

  1: Compute the Cholesky factorization $CC^\top = \mathbf{M}_d(\boldsymbol{\phi}^{*(d)}))$.

  2: Reduce $C$ to a column echelon form $U$.

  3: Compute from $U$ the multiplication matrices $N_j, \ j \in \{1, \ldots, p\}^2$.

  4: Set $N := \sum_{i=1}^{p} \kappa_i N_i$ with randomly generated coefficients $\kappa_i$.

  5: Compute the Schur decomposition $N = QTQ^\top$.

  6: Compute the column vectors $\{q_i\}_{1 \leq i \leq K}$ of $Q$.

  7: Return $\widehat{\theta}_{j,\text{dist},i} := q_i^\top N_j q_i, \ i \in \{1, \ldots, K\}, \ j \in \{1, \ldots, p\}$.

---

[2] Multiplication matrices $N_j$ capture how multiplication by $\theta_j$ acts on the polynomial basis generated by the moment matrix under the flat extension condition, where products of basis monomials are expressed modulo the kernel of the moment matrix. They can be computed by solving a linear system.

**Hahn-Jordan decomposition.** Given a *signed* measure $\phi$ on $S \in \mathcal{B}(\mathbb{R}^p)$, there always exist $P, N \in \mathcal{B}(\mathbb{R}^p)$ satisfying $P \cup N = S$ and $P \cap N = \emptyset$, and such that

$$\phi(E) \geq 0, \quad \forall E \in \mathcal{B}(\mathbb{R}^p), \, E \subseteq P \quad \text{and} \quad \phi(E) \leq 0, \quad \forall E \in \mathcal{B}(\mathbb{R}^p), \, E \subseteq N. \tag{22}$$

There also exists a *unique* decomposition of the form $\phi = \phi^+ - \phi^-$, with $\phi^+$ and $\phi^-$ being two positive measures, such that for every $E \in \mathcal{B}(\mathbb{R}^p)$:

$$\phi^+(E) = \phi(E \cap P), \quad \text{and} \quad \phi^-(E) = -\phi(E \cap N). \tag{23}$$

In addition, $\|\phi\|_{TV} = \phi^+(S) + \phi^-(S)$. In case where $\phi = \mu - \nu$ for two positive measure $\mu$ and $\nu$, we also have following measure domination relations: $\mu \geq \phi^+$ and $\nu \geq \phi^-$.

**Multivariate Carleman condition for mixture models.** In Lemma A.1, we establish a general result for mixtures of parametric probability distributions whose parameters lie in a compact basic semialgebraic set. This result is a key ingredient in our convergence proofs and constitutes an independent contribution.

**Lemma A.1** (Multivariate Carleman condition for mixture models). *Let $\nu_\phi \in \mathscr{P}(\mathbb{R}^n)$ be a mixture distribution with mixing measure $\phi \in \mathscr{P}(S_\theta)$, with $S_\theta \subset \mathbb{R}^p$ as in (4) compact. For every multi-index $\boldsymbol{\alpha} \in \mathbb{N}^n$, denote by $p_{\boldsymbol{\alpha}} \in \mathbb{R}_{|\boldsymbol{\alpha}|}[\boldsymbol{\theta}]$ the polynomial representation of the $\boldsymbol{\alpha}$-th moment of $\nu_\phi$ (i.e. $\int x^{\boldsymbol{\alpha}} d\nu_\phi = \int_{S_\theta} p_{\boldsymbol{\alpha}}(\boldsymbol{\theta}) d\phi \boldsymbol{\theta}$). Then, for each coordinate direction $\boldsymbol{e}_i \in \mathbb{N}^n$, $i = \{1, \ldots, n\}$, we have*

$$\sum_{k=1}^\infty \left( L_\phi(p_{2k\boldsymbol{e}_i}) \right)^{-\frac{1}{2k}} = +\infty, \tag{24}$$

*and therefore $\nu_\phi$ is moment-determinate.*

*Proof.* Moment determinacy follows directly from multivariate Carleman condition (24) and so it suffices to prove (24). Fix $i \in \{1, \ldots, n\}$. We proceed to bound even moments of $\nu_\phi$ along the direction $i$, for different parametric distribution classes with polynomial moments.

*Family I - Gaussian mixtures.* In this case, $\boldsymbol{\theta} = (\boldsymbol{m}, \boldsymbol{\Sigma})$, so that the $\pi_i \# \nu_\phi$ is distributed as $\mathcal{N}(m_i, \Sigma_{ii})$. By definition of Gaussian moments, for any $k \in \mathbb{N}$, we can write

$$p_{2k\boldsymbol{e}_i}(\boldsymbol{\theta}) = \sum_{l=0}^k \binom{2k}{2l} m_i^{2k-2l} \Sigma_{ii}^l (2l-1)!!, \tag{25}$$

where $(2l-1)!! = 1 \cdot 3 \cdot 5 \cdots (2l-1)$ denotes the double factorial of an odd integer, and $-1!! = 0!! = 1$. Since $S_\theta$ is compact, there exists $M > 0$ such that for all $\boldsymbol{\theta} \in S_\theta$, $\|\boldsymbol{\theta}\|_\infty \leq M$. This in turn implies that

$$|m_i| \leq M_{m,i}, \qquad 0 \leq \Sigma_{ii} \leq M_{\sigma,i}^2 \tag{26}$$

for some $M_{m,i}, M_{\sigma,i} > 0$.

Observe that $\pi_i \# \nu_\phi = m_i + \sqrt{\Sigma_{ii}} Z$ with $Z \sim \mathcal{N}(0,1)$, and that the moments of the standard Gaussian distribution are given by $z_{2k} = \frac{(2k)!}{2^k k!} \leq (2k)^k$. Moreover, we recall that $|a+b|^{2k} \leq 2^{2k-1}(|a|^{2k} + |b|^{2k})$ for all $a, b \in \mathbb{R}$ and $k \in \mathbb{N}$. Then, we can write

$$p_{2k\boldsymbol{e}_i}(\boldsymbol{\theta}) = \int_{\mathbb{R}} (m_i + \sqrt{\Sigma_{ii}}z)^{2k} dz \leq 2^{2k-1}\left( |m_i|^{2k} + \Sigma_{ii}^k z_{2k} \right)$$

$$\leq 2^{2k-1}\left( M_{m,i}^{2k} + \left( M_{\sigma,i}\sqrt{2k} \right)^{2k} \right) \tag{27}$$

$$\leq \left( C_{G,i}\sqrt{k} \right)^{2k}, \text{ with } C_{G,i} := 2\left( M_{m,i} + \sqrt{2}\, M_{\sigma,i} \right).$$

Therefore, by linearity and positivity of $L_\phi$,

$$L_\phi(p_{2k\boldsymbol{e}_i}) \leq \left( C_{G,i}\sqrt{k} \right)^{2k} \implies \sum_{k=1}^{+\infty} \left( L_\phi(p_{2k\boldsymbol{e}_i}) \right)^{-\frac{1}{2k}} \geq \sum_{k=1}^{+\infty} \frac{1}{C_{G,i}\sqrt{k}} = +\infty. \tag{28}$$

*Family II - Poisson mixtures.* In this case, the parameter is $\boldsymbol{\theta} = (\lambda_1, \ldots, \lambda_n) \in \mathbb{R}_{>0}^n$ with

the $i$-th marginal distributed as $\text{Pois}(\lambda_i)$. For any $k \in \mathbb{N}$, the (component) $2k$-th marginal moment is given by the Touchard polynomial $T_{2k}$, i.e.,

$$p_{2k e_i}(\boldsymbol{\theta}) = \int_{\mathbb{R}^n} x_i^{2k} \, d\nu_\phi(\boldsymbol{x}) = T_{2k}(\lambda_i) := \sum_{j=0}^{2k} S(2k, j) \, \lambda_i^j, \tag{29}$$

where $S(\cdot, \cdot)$ are Stirling numbers of the second kind. Since $S_{\boldsymbol{\theta}}$ is compact, there exists $\Lambda_i > 0$ such that $0 < \lambda_i \le \Lambda_i$. Using the inequality $T_{2k}(\lambda_i) \le B_{2k}(1 + \Lambda_i)^{2k}$, where the Bell numbers $B_m := \sum_{j=0}^m S(m, j)$ satisfy the standard bound $B_m \le \left( C_B \frac{m}{\ln(m+1)} \right)^m$ for all $m \in \mathbb{N}$ and some absolute constant $C_B > 0$, we obtain

$$p_{2k e_i}(\boldsymbol{\theta}) \le B_{2k}(1 + \Lambda_i)^{2k} \le \left( \frac{2 C_B (1 + \Lambda_i) k}{\ln(2k + 1)} \right)^{2k}. \tag{30}$$

Therefore, by linearity and positivity of $L_\phi$,

$$L_\phi\left(p_{2k e_i}\right) \le \left( \frac{k}{\widetilde{C} \ln(2k+1)} \right)^{2k} \implies \sum_{k=2}^{+\infty} \left( L_\phi(p_{2k e_i}) \right)^{-\frac{1}{2k}} \ge \sum_{k=2}^{+\infty} \frac{\widetilde{C} \ln(2k+1)}{k} = +\infty, \tag{31}$$

with $\widetilde{C} := \dfrac{1}{2 C_B (1 + \Lambda_i)}$.

*Family III - Exponential mixtures.* Here the distribution is controlled by $\boldsymbol{\theta} = (\lambda_1, \ldots, \lambda_n) \in \mathbb{R}_{>0}^n$ and the $i$-th marginal is $\text{Exp}(\lambda_i)$ supported on $\mathbb{R}_{>0}$. To obtain a *polynomial* moment map in the parameter, we set $\eta_i := \frac{1}{\lambda_i}$ with $\eta_i \in [\eta_{i,\min}, \eta_{i,\max}]$, which is possible since $S_{\boldsymbol{\theta}}$ is compact and $\lambda_i > 0$. For any $k \in \mathbb{N}$, the $2k$-th marginal moment is

$$p_{2k e_i}(\boldsymbol{\theta}) = \int_{\mathbb{R}^n} x_i^{2k} \, d\nu_\phi(\boldsymbol{x}) = \int_{\mathbb{R}_+} x^{2k} \lambda_i e^{-\lambda_i x} \, dx = (2k)! \, \eta_i^{2k}. \tag{32}$$

Using $(2k)! \le (2k)^{2k}$, we get

$$p_{2k e_i}(\boldsymbol{\theta}) \le (2k)^{2k} \eta_{i,\max}^{2k} = \left( 2 \, \eta_{i,\max} k \right)^{2k}. \tag{33}$$

Therefore, by linearity and positivity of $L_\phi$,

$$L_\phi\left(p_{2k e_i}\right) \le \left( 2 \, \eta_{i,\max} k \right)^{2k} \implies \sum_{k=1}^{+\infty} \left( L_\phi(p_{2k e_i}) \right)^{-\frac{1}{2k}} \ge \frac{1}{2 \, \eta_{i,\max}} \sum_{k=1}^{+\infty} \frac{1}{k} = +\infty. \tag{34}$$

$\square$

Lemma A.1 is crucial for our theoretical analysis and can also be shown to hold for other families of parametric densities, such as Binomial distributions, SOS-based densities, and others, thereby ensuring a broad applicability of our framework.

## A.3   CASE 1: WASSERSTEIN DISTANCE

We start by providing a fully detailed form of the optimization problem (12), namely:

$$\tau_{d,\varepsilon,R}^{\mathrm{W}_2} := \begin{cases} \displaystyle\inf_{\boldsymbol{\lambda} \in \mathbb{R}^{\mathbb{N}_{2d}^{2n}}, \boldsymbol{\phi} \in \mathbb{R}^{\mathbb{N}_{2d}^p}} L_{\boldsymbol{\lambda}}(\|\boldsymbol{x} - \boldsymbol{y}\|^2) + \varepsilon L_\phi(R) & \text{(35a)} \\[2mm] \text{s.t.} \quad \lambda_{\boldsymbol{\alpha},\mathbf{0}} - \mu_{\boldsymbol{\alpha}} = 0, \; \lambda_{\mathbf{0},\boldsymbol{\alpha}} - L_\phi(p_{\boldsymbol{\alpha}}) = 0, \; \boldsymbol{\alpha} \in \mathbb{N}_{2d}^n, & \text{(35b)} \\[2mm] \mathbf{M}_d(\boldsymbol{\lambda}) \succeq 0, & \text{(35c)} \\[2mm] \mathbf{M}_{d-d_j}(r_j \phi) \succeq 0, \; j \in \{0, \ldots, l\}. & \text{(35d)} \end{cases}$$

To establish an explicit correspondence between problem (12) and problem 35, let $\boldsymbol{\lambda} = (\lambda_{\boldsymbol{\alpha},\boldsymbol{\beta}})_{|\boldsymbol{\alpha}| + |\boldsymbol{\beta}| \le 2d}$ and $\boldsymbol{\phi} = (\phi_{\boldsymbol{\gamma}})_{|\boldsymbol{\gamma}| \le 2d}$ collect all pseudo-moments indexed by monomials in $(\boldsymbol{x}, \boldsymbol{y})$ and $\boldsymbol{\theta}$, respectively, up to total degree $2d$.

The objective $L_{\boldsymbol{\lambda}}(\|\boldsymbol{x}-\boldsymbol{y}\|^2) + \varepsilon L_{\boldsymbol{\phi}}(R)$ is linear in $(\boldsymbol{\lambda}, \boldsymbol{\phi})$, and if we write

$$\|\boldsymbol{x}-\boldsymbol{y}\|^2 = \sum_{|\boldsymbol{\alpha}|+|\boldsymbol{\beta}|\leq 2d} u_{\boldsymbol{\alpha},\boldsymbol{\beta}}\, \boldsymbol{x}^{\boldsymbol{\alpha}}\boldsymbol{y}^{\boldsymbol{\beta}},\ R(\boldsymbol{\theta}) = \sum_{|\boldsymbol{\gamma}|\leq 2d} v_{\boldsymbol{\gamma}}\, \boldsymbol{\theta}^{\boldsymbol{\gamma}}, \tag{36}$$

we obtain

$$L_{\boldsymbol{\lambda}}(\|\boldsymbol{x}-\boldsymbol{y}\|^2) + \varepsilon L_{\boldsymbol{\phi}}(R) = \sum_{|\boldsymbol{\alpha}|+|\boldsymbol{\beta}|\leq 2d} u_{\boldsymbol{\alpha},\boldsymbol{\beta}}\, \lambda_{\boldsymbol{\alpha},\boldsymbol{\beta}} + \varepsilon \sum_{|\boldsymbol{\gamma}|\leq 2d} v_{\boldsymbol{\gamma}}\, \phi_{\boldsymbol{\gamma}}. \tag{37}$$

Therefore, the cost vector in (12) is given by:

$$\mathbf{c}_{d,\varepsilon,R}^{\mathrm{W_2}} := \big((u_{\boldsymbol{\alpha},\boldsymbol{\beta}})_{|\boldsymbol{\alpha}|+|\boldsymbol{\beta}|\leq 2d},\ (\varepsilon v_{\boldsymbol{\gamma}})_{|\boldsymbol{\gamma}|\leq 2d}\big). \tag{38}$$

To make the linear operator in (12) explicit, let us explicitly enumerate $\mathbb{N}_{2d}^n = \{\boldsymbol{\alpha}^1, \ldots, \boldsymbol{\alpha}^{s(n,2d)}\}$, and define the truncated moment vector $\boldsymbol{\mu}_d := (\mu_{\boldsymbol{\alpha}^1}, \ldots, \mu_{\boldsymbol{\alpha}^{s(n,2d)}})^\top$.
We now introduce matrices $\mathbf{E}^{(1)}, \mathbf{E}^{(2)} \in \mathbb{R}^{s(n,2d)\times s(n,2d)}$ and $\mathbf{P} \in \mathbb{R}^{s(n,2d)\times s(p,2d)}$ such that:

$$(\mathbf{E}^{(1)}\boldsymbol{\lambda})_i = \lambda_{\boldsymbol{\alpha}^i,\mathbf{0}},\ (\mathbf{E}^{(2)}\boldsymbol{\lambda})_i = \lambda_{\mathbf{0},\boldsymbol{\alpha}^i},\ (\mathbf{P}\boldsymbol{\phi})_i = \sum_{|\boldsymbol{\gamma}|\leq 2d} p_{\boldsymbol{\alpha}^i,\boldsymbol{\gamma}}\, \phi_{\boldsymbol{\gamma}},\ i\in\{1,\ldots,s(n,2d)\}, \tag{39}$$

where $p_{\boldsymbol{\alpha}^i}(\boldsymbol{\theta}) = \sum_{|\boldsymbol{\gamma}|\leq 2d} p_{\boldsymbol{\alpha}^i,\boldsymbol{\gamma}}\, \boldsymbol{\theta}^{\boldsymbol{\gamma}}$. Then the moment-matching constraints from (35b) can be enforced using :

$$\mathbf{A}_d^{\mathrm{W_2}} := \begin{bmatrix} \mathbf{E}^{(1)} & \mathbf{0}_{s(n,2d)\times s(n,2d)} \\ \mathbf{E}^{(2)} & -\mathbf{P} \end{bmatrix}, \qquad \mathbf{b}_d^{\mathrm{W_2}} := \begin{bmatrix} \boldsymbol{\mu} \\ \mathbf{0}_{s(n,2d)} \end{bmatrix}. \tag{40}$$

Finally, the semidefinite constraints (35c)–(35d) are written compactly via:

$$\mathbf{M}_d^{\mathrm{W_2}}(\boldsymbol{y}_d^{\mathrm{W_2}}) := \begin{pmatrix} \mathbf{M}_d(\boldsymbol{\lambda}) & \mathbf{0} & \cdots & \mathbf{0} \\ \mathbf{0} & \mathbf{M}_{d-d_0}(r_0\boldsymbol{\phi}) & \cdots & \mathbf{0} \\ \vdots & \vdots & \ddots & \vdots \\ \mathbf{0} & \mathbf{0} & \cdots & \mathbf{M}_{d-d_l}(r_l\boldsymbol{\phi}) \end{pmatrix} \succeq 0, \tag{41}$$

where $\mathbf{M}_d(\boldsymbol{\lambda})$ is the moment matrix of order $d$ associated with $\boldsymbol{\lambda}$ and $\mathbf{M}_{d-d_j}(r_j\boldsymbol{\phi})$ are the corresponding localizing matrices.

Let us now derive the SDP dual (SOS relaxation) of the problem presented in (35). Consider $q \in \mathbb{R}_{2d}[\boldsymbol{x}]$ and $g \in \mathbb{R}_{2d}[\boldsymbol{y}]$. The associated Lagrangian can be written as:

$$\begin{aligned}
\mathcal{L}(\boldsymbol{\lambda},\boldsymbol{\phi};q,g) := {}& L_{\boldsymbol{\lambda}}(\|\boldsymbol{x}-\boldsymbol{y}\|^2) + \varepsilon L_{\boldsymbol{\phi}}(R) \\
& + \sum_{\boldsymbol{\alpha}\in\mathbb{N}_{2d}^n} q_{\boldsymbol{\alpha}}(\mu_{\boldsymbol{\alpha}} - \lambda_{\boldsymbol{\alpha},\mathbf{0}}) + \sum_{\boldsymbol{\alpha}\in\mathbb{N}_{2d}^n} g_{\boldsymbol{\alpha}}(\lambda_{\mathbf{0},\boldsymbol{\alpha}} - L_{\boldsymbol{\phi}}(p_{\boldsymbol{\alpha}})) \\
= {}& \sum_{\boldsymbol{\alpha}\in\mathbb{N}_{2d}^n} q_{\boldsymbol{\alpha}}\mu_{\boldsymbol{\alpha}} + L_{\boldsymbol{\lambda}}\left(\|\boldsymbol{x}-\boldsymbol{y}\|^2 - \sum_{\boldsymbol{\alpha}\in\mathbb{N}_{2d}^n} q_{\boldsymbol{\alpha}}\boldsymbol{x}^{\boldsymbol{\alpha}} + \sum_{\boldsymbol{\alpha}\in\mathbb{N}_{2d}^n} g_{\boldsymbol{\alpha}}\boldsymbol{y}^{\boldsymbol{\alpha}}\right) \\
& + L_{\boldsymbol{\phi}}\left(\varepsilon R - \sum_{\boldsymbol{\alpha}\in\mathbb{N}_{2d}^n} g_{\boldsymbol{\alpha}}p_{\boldsymbol{\alpha}}\right)
\end{aligned} \tag{42}$$

From (35c)-(35d), we deduce that the arguments of $L_{\boldsymbol{\lambda}}$ and $L_{\boldsymbol{\phi}}$ (last two terms of (42)) must be nonnegative over appropriate domains (supports). Finally, the SDP dual of (35) corresponds to:

$$\tau_{d,\varepsilon,R}^{\mathrm{W_2},*} := \begin{cases} \displaystyle\sup_{q\in\mathbb{R}_{2d}[\boldsymbol{x}],g\in\mathbb{R}_{2d}[\boldsymbol{y}]} \sum_{\boldsymbol{\alpha}\in\mathbb{N}_{2d}^n} q_{\boldsymbol{\alpha}}\mu_{\boldsymbol{\alpha}} & \text{(43a)} \\[6pt] \text{s.t.} \quad h - q + g \in \Sigma_d[\boldsymbol{x},\boldsymbol{y}],\ h:(\boldsymbol{x},\boldsymbol{y})\mapsto \|\boldsymbol{x}-\boldsymbol{y}\|^2, & \text{(43b)} \\[6pt] \displaystyle \varepsilon R - \sum_{\boldsymbol{\alpha}\in\mathbb{N}_{2d}^n} g_{\boldsymbol{\alpha}}p_{\boldsymbol{\alpha}} \in \mathcal{Q}_d(r) \subset \Sigma_d[\boldsymbol{\theta}]. & \text{(43c)} \end{cases}$$

By standard results in SDP duality theory, in particular the existence of strictly feasible points, one can show that, up to numerical inaccuracies, there is no duality gap. Hence, for all $d \geq d_{\min}$, $\tau_{d,\varepsilon,R}^{W_2} = \tau_{d,\varepsilon,R}^{W_2,*}$.

We now establish the convergence of our regularized hierarchy by adapting the proof of (Lasserre, 2024a) to the multivariate setting and to more general families of mixtures that are moment-determinate under the conditions stated, for example, in Proposition A.1.

**Theorem A.1** ($W_2$ case: existence of optimal solutions and convergence). *Let $S_{\boldsymbol{\theta}}$ be as in (4) compact and $\mu \in \mathscr{P}(\mathbb{R}^n)$ satisfying Assumptions 2.1 and 2.2. For any $d \geq d_{\min}$, problem (35) admits an optimal solution $(\boldsymbol{\lambda}^{*(d)}, \boldsymbol{\phi}^{*(d)})$. Moreover, any accumulation point $(\boldsymbol{\lambda}^*, \boldsymbol{\phi}^*)$ of the sequence $(\boldsymbol{\lambda}^{*(d)}, \boldsymbol{\phi}^{*(d)})_{d \geq d_{\min}}$ solves (10), implying that $\lim_{d \to +\infty} \tau_{d,\varepsilon,R}^{W_2} = \tau_{\varepsilon,R}^{W_2}$.*

*Proof.*

*Part I: Existence of optimal solutions.* Let $d \geq d_{\min}$ and $(\boldsymbol{\lambda}, \boldsymbol{\phi})$ be feasible for (35). Since $S_{\boldsymbol{\theta}}$ is compact, we might assume that $r_{l+1}(\boldsymbol{\theta}) := R^2 - \|\boldsymbol{\theta}\|^2$ is a redundant constraint for the feasible set of mixture parameters, for some $R > 0$ large enough. The associated localizing constraint $\mathbf{M}_{d-1}(r_{l+1}\phi) \succeq 0$ would then imply that $L_{\boldsymbol{\phi}}(\theta_j^{2d}) < R^{2d}$, for all $j \in \{1, \dots, p\}$. Then, combing the fact that $\mathbf{M}_d(\boldsymbol{\phi}) \succeq 0$ with (Lasserre, 2015, Proposition 2.38, p 41), we deduce that

$$|\phi_{\boldsymbol{\alpha}}| \leq \max\{1, R^{2d}\} \quad \text{for all} \quad 0 \neq \boldsymbol{\alpha} \in \mathbb{N}_{2d}^p. \tag{44}$$

Furthermore, by Assumption 2.2, $p_{\boldsymbol{\alpha}} \in \mathbb{R}_{|\boldsymbol{\alpha}|}[\boldsymbol{\theta}]$, so we can write $p_{\boldsymbol{\alpha}}(\boldsymbol{\theta}) = \sum_{\boldsymbol{\gamma} \in \mathbb{N}_{|\boldsymbol{\alpha}|}^p} a_{\boldsymbol{\gamma}} \boldsymbol{\theta}^{\boldsymbol{\gamma}}$, for some $a_{\boldsymbol{\gamma}} \in \mathbb{R}$. This, combined with the compactness of $S_{\boldsymbol{\theta}}$ yields

$$|p_{\boldsymbol{\alpha}}(\boldsymbol{\theta})| \leq \sum_{\boldsymbol{\gamma} \in \mathbb{N}_{|\boldsymbol{\alpha}|}^p} |a_{\boldsymbol{\gamma}}| |\boldsymbol{\theta}^{\boldsymbol{\gamma}}| \leq C \widetilde{R}^{|\boldsymbol{\alpha}|}, \ \ \widetilde{R} := 1 + R, \ \ C := \sum_{\boldsymbol{\gamma} \in \mathbb{N}_{|\boldsymbol{\alpha}|}^p} |a_{\boldsymbol{\gamma}}|. \tag{45}$$

By linearity of the Riesz operator, we obtain $L_{\boldsymbol{\phi}}(p_{2\boldsymbol{\alpha}}) \leq C\widetilde{R}^{2|\boldsymbol{\alpha}|} L_{\boldsymbol{\phi}}(1) \leq C\widetilde{R}^{2d}$ for all $\boldsymbol{\alpha} \in \mathbb{N}_d^n$. Now, notice that $\lambda_{2\boldsymbol{\alpha},\mathbf{0}} = \mu_{2\boldsymbol{\alpha}} \leq (1 + \max_{\boldsymbol{\beta} \in \mathbb{N}_d^n} |\mu_{2\boldsymbol{\beta}}|)^{2d} := \overline{R}^{2d}$ and that $\lambda_{\mathbf{0},2\boldsymbol{\beta}} = L_{\boldsymbol{\phi}}(p_{2\boldsymbol{\beta}}) \leq C\widetilde{R}^{2d}$. Combining these observations with $\mathbf{M}_d(\boldsymbol{\lambda}) \succeq 0$, we can again apply (Lasserre, 2015, Proposition 2.38, p 41) to conclude that

$$\forall \boldsymbol{\alpha}, \boldsymbol{\beta} \in \mathbb{N}_d^n, \quad |\lambda_{\boldsymbol{\alpha},\boldsymbol{\beta}}| \leq \max\{1, \lambda_{2\boldsymbol{\alpha},\mathbf{0}}, \lambda_{\mathbf{0},2\boldsymbol{\beta}}\} \leq \max\{1, C\widetilde{R}^{2d}, \overline{R}^{2d}\}. \tag{46}$$

Equations (44) and (46) then imply that the feasible set of the SDP problem (35) is compact, and thus, by continuity, an optimal solution $(\boldsymbol{\lambda}^{*(d)}, \boldsymbol{\phi}^{*(d)}) \in \mathbb{N}_{2d}^{2n} \times \mathbb{N}_{2d}^p$ exists.

*Part II: Convergence of the hierarchy.* Any optimal solution $(\boldsymbol{\lambda}^{*(d)}, \boldsymbol{\phi}^{*(d)})$ of (35), can be extended into an infinite dimensional sequence by setting $\lambda_{\boldsymbol{\alpha},\boldsymbol{\beta}}^{*(d)} = 0$ whenever $|\boldsymbol{\alpha}| + |\boldsymbol{\beta}| > 2d$, and $\phi_{\boldsymbol{\gamma}}^{*(d)} = 0$ for $|\boldsymbol{\gamma}| > 2d$. Next, construct a normalized sequence $(\widetilde{\boldsymbol{\lambda}}^{(d)}, \widetilde{\boldsymbol{\phi}}^{(d)})$ by setting

$$\widetilde{\lambda}_{\boldsymbol{\alpha},\boldsymbol{\beta}}^{(d)} := \frac{\lambda_{\boldsymbol{\alpha},\boldsymbol{\beta}}^{*(d)}}{\max\{1, C\widetilde{R}^{2k}, \overline{R}^{2k}\}} \quad \text{and} \quad \widetilde{\phi}_{\boldsymbol{\gamma}}^{(d)} := \frac{\phi_{\boldsymbol{\gamma}}^{*(d)}}{\max\{1, R^{2k}\}}, \tag{47}$$

whenever $2k-1 \leq |\boldsymbol{\alpha} + \boldsymbol{\beta}| \leq 2k$ and $2k-1 \leq |\boldsymbol{\gamma}| \leq 2k$, $k = 1, \dots, d$. Such normalization enables us to conclude that both $\widetilde{\boldsymbol{\lambda}}^{(d)}$ and $\widetilde{\boldsymbol{\phi}}^{(d)}$ are elements of unit balls of appropriate Banach spaces of uniformly bounded sequences. By Banach–Alaoglu theorem, these unit balls are sequentially compact in weak-$\star$ topology, so we can always find a subsequence $(d_k)_{k \in \mathbb{N}}$ and infinite dimensional vectors $\widetilde{\boldsymbol{\lambda}}^*$ and $\widetilde{\boldsymbol{\phi}}^*$ satisfying

$$\lim_{k \to +\infty} \widetilde{\lambda}_{\boldsymbol{\alpha},\boldsymbol{\beta}}^{(d_k)} = \widetilde{\lambda}_{\boldsymbol{\alpha},\boldsymbol{\beta}}^*, \ \forall (\boldsymbol{\alpha}, \boldsymbol{\beta}) \in \mathbb{N}^n \times \mathbb{N}^n, \ \text{and} \ \lim_{k \to +\infty} \widetilde{\phi}_{\boldsymbol{\gamma}}^{(d_k)} = \widetilde{\phi}_{\boldsymbol{\gamma}}^*, \ \forall \boldsymbol{\gamma} \in \mathbb{N}^p. \tag{48}$$

Then the reverse scaling of the one in (47), applied to $\widetilde{\boldsymbol{\lambda}}^*$ and $\widetilde{\boldsymbol{\phi}}^*$, provides an infinite dimensional accumulation point $(\boldsymbol{\lambda}^*, \boldsymbol{\phi}^*)$.

Fix $d \geq d_{\min}$ and $j \in \{0, \dots, l+1\}$, arbitrary. By the entry-wise convergence (48) one obtains

$\mathbf{M}_d(\boldsymbol{\lambda}^*) \succeq 0$ and $\mathbf{M}_{d-d_j}(r_j \boldsymbol{\phi}^*) \succeq 0$. Moreover, again by (48), the moment matching constraints are also satisfied, i.e., $\lambda^*_{\boldsymbol{\alpha},\mathbf{0}} = \mu_{\boldsymbol{\alpha}}$, and $\lambda^*_{\mathbf{0},\boldsymbol{\beta}} = L_{\boldsymbol{\phi}^*}(p_{\boldsymbol{\beta}})$, for all $\boldsymbol{\alpha}, \boldsymbol{\beta} \in \mathbb{N}^n$.

It remains to prove that infinite dimensional vectors $(\boldsymbol{\lambda}^*, \boldsymbol{\phi}^*)$ admit a representing measure.

For $\boldsymbol{\phi}^*$, it follows from Putinar Positivstellensatz (Putinar, 1993) because the quadratic module $\mathcal{Q}_d(r)$ is Archimedean (thanks to the presence of the ball constraint $r_{l+1}$).

For $\boldsymbol{\lambda}^*$, observe that $L_{\boldsymbol{\lambda}^*}(x_i^{2k}) = \lambda^*_{2k\boldsymbol{e}_i,\mathbf{0}} = \mu_{2k\boldsymbol{e}_i}$, with $\boldsymbol{e}_i$ being the $i$-th unit vector in $\mathbb{N}^n$, and recall that $\mu$ satisfies Assumption 2.1. invoking Lemma A.1, it follows that the series whose general term is $(L_{\boldsymbol{\lambda}^*}(y_i^{2k}))^{-\frac{1}{2k}} = (\lambda^*_{\mathbf{0},2k\boldsymbol{e}_i})^{-\frac{1}{2k}} = (L_{\boldsymbol{\phi}^*}(p_{2k\boldsymbol{e}_i}))^{-\frac{1}{2k}}$ is divergent. Therefore the infinite vector $\boldsymbol{\lambda}^*$ satisfies the (multivariate) Carleman condition, and so according to (Lasserre, 2009, Theorem 3.5), $\boldsymbol{\phi}^*$ and $\boldsymbol{\lambda}^*$ admit moment determinate representing measures $\phi^* \in \mathscr{P}(S_{\boldsymbol{\theta}})$ and $\lambda^* \in \mathscr{P}(\mathbb{R}^n \times \mathbb{R}^n)$, respectively. This implies that the couple $(\lambda^*, \phi^*)$ is feasible in (10). Finally, using (48) and the fact that (35) is a relaxation, we deduce that

$$\tau^{\mathrm{W}_2}_{\varepsilon,R} \leq L_{\boldsymbol{\lambda}^*}(||\boldsymbol{x} - \boldsymbol{y}||^2) + \varepsilon L_{\boldsymbol{\phi}^*}(R) = \lim_{k \to +\infty} \tau^{\mathrm{W}_2}_{d_k,\varepsilon,R} \leq \tau^{\mathrm{W}_2}_{\varepsilon,R}, \tag{49}$$

proving the optimality of $(\lambda^*, \phi^*)$ in problem (10). $\qquad\square$

The proof of our theoretical result that can be used for unsupervised machine learning applications is now quite straightforward.

*Proof of Theorem (3.2).* Thanks to Theorem A.1, we know that the optimal solution $(\boldsymbol{\lambda}^{*(d)}, \boldsymbol{\phi}^{*(d)})$ exists for any $d \geq d_{\min}$. If $\operatorname{rank} \mathbf{M}_d(\boldsymbol{\phi}^{*(d)}) = \operatorname{rank} \mathbf{M}_{d-d_{\min}}(\boldsymbol{\phi}^{*(d)}) = K$, for $K \in \mathbb{N}$, then, the Curto-Fialkow flat extension theorem (Lasserre, 2015, Theorem 2.47) guarantees that $\boldsymbol{\phi}^{*(d)}$ is the truncated moment sequence of some $K-$atomic probability measure $\phi^* \in \mathscr{P}(S_{\boldsymbol{\theta}})$, i.e., there exist $\boldsymbol{\theta}_1, \ldots, \boldsymbol{\theta}_K \in S_{\boldsymbol{\theta}}$ and $\alpha_1, \ldots, a_K \in \mathbb{R}_{>0}$ satisfying $\sum_{j=1}^K \alpha_j = 1$ and $\phi^* = \sum_{j=1}^K \alpha_j \delta_{\boldsymbol{\theta}_j}$. Since, by Theorem A.1, such $\phi^*$ solves (10), it therefore represents the finite mixture that best approximates $\mu$ in $\mathrm{W}_2$-regularized sense. The parameter $K$ can be interpreted as the mixture order, points $\boldsymbol{\theta}_j$ (that can be extracted using Algorithm 2) parametrize each component, and values $\alpha_j$ are nothing but weights associated to each component. $\qquad\square$

## A.4 CASE 2 - TOTAL VARIATION

As in the previous subsection, we start by providing a fully detailed formulation of the optimization problem (17), namely:

$$\tau^{\mathrm{TV}}_{d,\varepsilon,R} := \begin{cases} \displaystyle\inf_{\boldsymbol{\psi}_+, \boldsymbol{\psi}_- \in \mathbb{R}^{\mathbb{N}^n_{2d}}, \boldsymbol{\phi} \in \mathbb{R}^{\mathbb{N}^p_{2d}}} \psi_{+,\mathbf{0}} + \psi_{-,\mathbf{0}} + \varepsilon L_{\boldsymbol{\phi}}(R) & \text{(50a)} \\[2mm] \text{s.t.} \quad \psi_{+,\boldsymbol{\alpha}} - \psi_{-,\boldsymbol{\alpha}} - \mu_{\boldsymbol{\alpha}} + L_{\boldsymbol{\phi}}(p_{\boldsymbol{\alpha}}) = 0, \ \boldsymbol{\alpha} \in \mathbb{N}^n_{2d}, & \text{(50b)} \\[1mm] \mathbf{M}_d(\boldsymbol{\mu}) \succeq \mathbf{M}_d(\boldsymbol{\psi}_+) \succeq 0, & \text{(50c)} \\[1mm] \mathbf{M}_d(p; \boldsymbol{\phi}) \succeq \mathbf{M}_d(\boldsymbol{\psi}_-) \succeq 0, & \text{(50d)} \\[1mm] \mathbf{M}_{d-d_j}(r_j \boldsymbol{\phi}) \succeq 0, \ j \in \{0, \ldots, l\}. & \text{(50e)} \end{cases}$$

To establish an explicit correspondence between problems (17) and (50), $\boldsymbol{\psi}_+ = (\psi^+_{\boldsymbol{\alpha}})_{|\boldsymbol{\alpha}| \leq 2d}$, $\boldsymbol{\psi}_- = (\psi^-_{\boldsymbol{\alpha}})_{|\boldsymbol{\alpha}| \leq 2d}$ and $\boldsymbol{\phi} = (\phi_{\boldsymbol{\gamma}})_{|\boldsymbol{\gamma}| \leq 2d}$ collect all pseudo-moments indexed by monomials in $\boldsymbol{x}$ and $\boldsymbol{\theta}$ up to degree $2d$, and set $\boldsymbol{y}^{\mathrm{TV}}_d := (\boldsymbol{\psi}_+, \boldsymbol{\psi}_-, \boldsymbol{\phi})$. The objective function in (50) is linear in $(\boldsymbol{\psi}_+, \boldsymbol{\psi}_-, \boldsymbol{\phi})$, and, following the same notation as in the $\mathrm{W}_2$ case, we deduce that the exact cost vector corresponds to:

$$\mathbf{c}^{\mathrm{TV}}_{d,\varepsilon,R} := (\mathbf{e_0}, \ \mathbf{e_0}, \ (\varepsilon v_{\boldsymbol{\gamma}})_{|\boldsymbol{\gamma}| \leq 2d}), \tag{51}$$

where $\mathbf{e_0} \in \mathbb{R}^{s(n,2d)}$ denotes the canonical basis vector corresponding to the index $\boldsymbol{\alpha} = \mathbf{0}$ in the coordinates of $\boldsymbol{\psi}_+$ and $\boldsymbol{\psi}_-$.

Recall that $\mathbf{I}_{s(n,2d)}$ stands for the identity matrix of size $s(n, 2d)$, and let $\mathbf{P} \in \mathbb{R}^{s(n,2d) \times s(p,2d)}$ be the same as in the previous subsection. Then the moment-matching constraints in (50b), can be written compactly using

$$\mathbf{A}^{\mathrm{TV}}_d := \begin{bmatrix} \mathbf{I}_d & -\mathbf{I}_d & \mathbf{P} \end{bmatrix}, \quad \text{and} \quad \mathbf{b}^{\mathrm{TV}}_d = \boldsymbol{\mu}_d. \tag{52}$$

Finally, let us introduce the following symmetric matrices:

$$\mathbf{S}_d^{\mathrm{TV}}(\boldsymbol{y}_d^{\mathrm{TV}}) := \begin{pmatrix} \mathbf{M}_d(\boldsymbol{\psi}_+) & \mathbf{0} & \mathbf{0} & \mathbf{0} \\ \mathbf{0} & \mathbf{M}_d(\boldsymbol{\mu}) - \mathbf{M}_d(\boldsymbol{\psi}_+) & \mathbf{0} & \mathbf{0} \\ \mathbf{0} & \mathbf{0} & \mathbf{M}_d(\boldsymbol{\psi}_-) & \mathbf{0} \\ \mathbf{0} & \mathbf{0} & \mathbf{0} & \mathbf{M}_d(p;\boldsymbol{\phi}) - \mathbf{M}_d(\boldsymbol{\psi}_-) \end{pmatrix}, \quad (53)$$

and

$$\mathbf{R}_d^{\mathrm{TV}}(\boldsymbol{y}_d^{\mathrm{TV}}) := \begin{pmatrix} \mathbf{M}_{d-d_0}(r_0\boldsymbol{\phi}) & \mathbf{0} & \cdots & \mathbf{0} \\ \mathbf{0} & \mathbf{M}_{d-d_1}(r_1\boldsymbol{\phi}) & \cdots & \mathbf{0} \\ \vdots & \vdots & \ddots & \vdots \\ 0 & 0 & \cdots & \mathbf{M}_{d-d_l}(r_l\boldsymbol{\phi}) \end{pmatrix}. \quad (54)$$

Then, the semidefinite constraints (50c)–(50e) can be compactly written as:

$$\mathbf{M}_d^{\mathrm{TV}}(\boldsymbol{y}_d^{\mathrm{TV}}) := \begin{pmatrix} \mathbf{S}_d^{\mathrm{TV}}(\boldsymbol{y}_d^{\mathrm{TV}}) & \mathbf{0} \\ \mathbf{0} & \mathbf{R}_d^{\mathrm{TV}}(\boldsymbol{y}_d^{\mathrm{TV}}) \end{pmatrix} \succeq 0, \quad (55)$$

which is equivalent to requiring each diagonal block in $\mathbf{S}_d^{\mathrm{TV}}(\boldsymbol{y}_d^{\mathrm{TV}})$ and $\mathbf{R}_d(\boldsymbol{y}_d^{\mathrm{TV}})$ to be positive semidefinite. All off-diagonal blocks are zero matrices of appropriate sizes.

To derive the SDP dual of (50), observe that the Lagrange multipliers corresponding to the SDP constraints (50c)–(50d) are positive semidefinite matrices. These can be interpreted as the Gram matrices of certain sum-of-squares (SOS) polynomials. Therefore, let $\sigma_+, \sigma_- \in \Sigma_d[\boldsymbol{x}]$ and $q \in \mathbb{R}_{2d}[\boldsymbol{x}]$. The associated Lagrangian is then given by:

$$\begin{aligned}
\mathcal{L}(\boldsymbol{\psi}_\pm, \boldsymbol{\phi}; q, \sigma_\pm) &= \psi_{+,\mathbf{0}} + \psi_{-,\mathbf{0}} + \varepsilon L_{\boldsymbol{\phi}}(R) \\
&\quad + \sum_{\boldsymbol{\alpha} \in \mathbb{N}_{2d}^n} q_{\boldsymbol{\alpha}} \left[ \psi_{+,\boldsymbol{\alpha}} - \psi_{-,\boldsymbol{\alpha}} - \mu_{\boldsymbol{\alpha}} + L_{\boldsymbol{\phi}}(p_{\boldsymbol{\alpha}}) \right] \\
&\quad + \langle \Sigma_+, \mathbf{M}_d(\boldsymbol{\psi}_+) - \mathbf{M}_d(\boldsymbol{\mu}) \rangle + \langle \Sigma_-, \mathbf{M}_d(\boldsymbol{\psi}_-) - \mathbf{M}_d(p;\boldsymbol{\phi}) \rangle \\
&= -\sum_{\boldsymbol{\alpha} \in \mathbb{N}_{2d}^n} q_{\boldsymbol{\alpha}} \mu_{\boldsymbol{\alpha}} + L_{\boldsymbol{\psi}_+}(1+q) + L_{\boldsymbol{\psi}_-}(1-q) \\
&\quad + L_{\boldsymbol{\phi}} \left( \varepsilon R + \sum_{\boldsymbol{\alpha} \in \mathbb{N}_{2d}^n} q_{\boldsymbol{\alpha}} p_{\boldsymbol{\alpha}} \right) \\
&\quad + \langle \Sigma_+, \mathbf{M}_d(\boldsymbol{\psi}_+) \rangle^3 + \langle \Sigma_-, \mathbf{M}_d(\boldsymbol{\psi}_-) \rangle \\
&\quad - \langle \Sigma_+, \mathbf{M}_d(\boldsymbol{\mu}) \rangle - \langle \Sigma_-, \mathbf{M}_d(p;\boldsymbol{\phi}) \rangle \\
&= \sum_{\boldsymbol{\alpha} \in \mathbb{N}_{2d}^n} (-q_{\boldsymbol{\alpha}} - \sigma_{+,\boldsymbol{\alpha}}) \mu_{\boldsymbol{\alpha}} + L_{\boldsymbol{\psi}_+} \underbrace{(1 + q + \sigma_+)}_{(a)} + L_{\boldsymbol{\psi}_-} \underbrace{(1 - q + \sigma_-)}_{(b)} \\
&\quad + L_{\boldsymbol{\phi}} \left( \underbrace{\varepsilon R + \sum_{\boldsymbol{\alpha} \in \mathbb{N}_{2d}^n} (q_{\boldsymbol{\alpha}} - \sigma_{-,\boldsymbol{\alpha}}) p_{\boldsymbol{\alpha}}}_{(c)} \right).
\end{aligned} \quad (56)$$

By duality, and to ensure the finiteness of $\inf_{\boldsymbol{\psi}_\pm, \boldsymbol{\phi}} \mathcal{L}(\boldsymbol{\psi}_\pm, \boldsymbol{\phi}; q, \sigma_\pm)$, polynomial expressions in (a), (b) and (c) must not take negative values, which can be achieved by requiring them to have

adequate SOS decompositions.

Finally, we derive the dual problem corresponding to the problem (50) :

$$
\tau_{d,\varepsilon,R}^{\mathrm{TV},*} := \begin{cases}
\displaystyle\sup_{q\in\mathbb{R}_{2d}[\boldsymbol{x}],\sigma_\pm\in\Sigma_d[\boldsymbol{x}]} \sum_{\boldsymbol{\alpha}\in\mathbb{N}_{2d}^n} (-q_{\boldsymbol{\alpha}}-\sigma_{+,\boldsymbol{\alpha}})\mu_{\boldsymbol{\alpha}} & \text{(57a)} \\[2ex]
\text{s.t.} \quad 1+q+\sigma_+ \in \Sigma_d[\boldsymbol{x}], & \text{(57b)} \\[1ex]
\qquad\quad 1-q+\sigma_- \in \Sigma_d[\boldsymbol{x}], & \text{(57c)} \\[1ex]
\qquad\quad \varepsilon R + \displaystyle\sum_{\boldsymbol{\alpha}\in\mathbb{N}_{2d}^n}(q_{\boldsymbol{\alpha}}-\sigma_{-,\boldsymbol{\alpha}})p_{\boldsymbol{\alpha}} \in \mathcal{Q}_d(r). & \text{(57d)}
\end{cases}
$$

One can prove that Slater's condition holds for the pair of SDP problems (50)-(57), implying that there is no duality gap (up to numerical inaccuracies). Therefore, for all $d \geq d_{\min}$, $\tau_{d,\varepsilon,R}^{\mathrm{TV}} = \tau_{d,\varepsilon,R}^{\mathrm{TV},*}$. Like in the $W_2$ case, let us now prove the convergence of our regularized hierarchy for computing the best TV-approximation of a mixture known only through a finite number of its moments.

**Theorem A.2** (TV case: existence of optimal solutions and convergence). *Let $S_{\boldsymbol{\theta}}$ be as in (4) compact and $\mu \in \mathscr{P}(\mathbb{R}^n)$ satisfying Assumptions 2.1 and 2.2. For any $d \geq d_{\min}$, problem (50) admits an optimal solution $(\boldsymbol{\psi}_+^{*(d)}, \boldsymbol{\psi}_-^{*(d)}, \boldsymbol{\phi}^{*(d)})$. Moreover, any accumulation point $(\boldsymbol{\psi}_+^*, \boldsymbol{\psi}_-^*, \boldsymbol{\phi}^*)$ of the sequence of optimal pseudo-moment vectors $(\boldsymbol{\psi}_+^{*(d)}, \boldsymbol{\psi}_-^{*(d)}, \boldsymbol{\phi}^{*(d)})_{d\geq d_{\min}}$ solves (16), implying $\lim_{d\to+\infty} \tau_{d,\varepsilon,R}^{\mathrm{TV}} = \tau_{\varepsilon,R}^{\mathrm{TV}}$.*

*Proof.* The proof mimicks that of Theorem A.1.

*Part I: Existence of optimal solutions.* Let $(\boldsymbol{\psi}_+, \boldsymbol{\psi}_-, \boldsymbol{\phi})$ be feasible for (50). Notice that for all $0 \neq \boldsymbol{\alpha} \in \mathbb{N}_{2d}^p$, $|\phi_{\boldsymbol{\alpha}}|$ can be bounded the same way as in (44). Furthermore, observe that

$$
\mathbf{M}_d(\boldsymbol{\mu}) - \mathbf{M}_d(\boldsymbol{\psi}_+) \succeq 0 \iff \boldsymbol{q}^\top(\mathbf{M}_d(\boldsymbol{\mu}) - \mathbf{M}_d(\boldsymbol{\psi}_+))\boldsymbol{q} \geq 0, \ \forall \boldsymbol{q} \in \mathbb{R}^{\binom{n+d}{d}}, \tag{58}
$$

which, by using the definition of Riesz functional can be rewritten as $L_{\boldsymbol{\mu}}(q^2) - L_{\boldsymbol{\psi}_+}(q^2) \geq 0$ for all $q \in \mathbb{R}_d[\boldsymbol{x}]$. In particular, we obtain

$$
\psi_{+,\mathbf{0}} \leq \mu_\mathbf{0} \quad \text{and} \quad L_{\boldsymbol{\psi}_+}(x_i^{2d}) \leq L_{\boldsymbol{\mu}}(x_i^{2d}), \ \forall i \in \{1,\dots,n\}. \tag{59}
$$

Since, additionally, $\mathbf{M}_d(\boldsymbol{\psi}_+) \succeq 0$, by using (Lasserre, 2015, Proposition 2.38, p 41), we get

$$
\forall \boldsymbol{\alpha} \in \mathbb{N}_{2d}^n, \ |\psi_{+,\boldsymbol{\alpha}}| \leq \max\left\{\mu_\mathbf{0}, \max_{i\in\{1,\dots,n\}} L_{\boldsymbol{\mu}}(x_i^{2d})\right\}. \tag{60}
$$

Similarly, as $\mathbf{M}_d(p;\boldsymbol{\phi}) - \mathbf{M}_d(\boldsymbol{\psi}_-) \succeq 0$, we deduce that $L_{\boldsymbol{\phi}}\left(\displaystyle\sum_{\boldsymbol{\alpha},\boldsymbol{\beta}\in\mathbb{N}_d^n} q_{\boldsymbol{\alpha}}q_{\boldsymbol{\beta}}p_{\boldsymbol{\alpha}+\boldsymbol{\beta}}\right) - L_{\boldsymbol{\psi}_-}(q^2) \geq 0$

for all $q \in \mathbb{R}_d[\boldsymbol{x}]$, and in particular

$$
\psi_{-,\mathbf{0}} \leq \phi_\mathbf{0} \quad \text{and} \quad L_{\boldsymbol{\psi}_-}(x_i^{2d}) \leq L_{\boldsymbol{\phi}}(p_{2d\boldsymbol{e}_i}), \ \forall i \in \{1,\dots,n\}. \tag{61}
$$

Positive semidefiniteness of $\mathbf{M}_d(\boldsymbol{\psi}_-)$ then implies

$$
\forall \boldsymbol{\alpha} \in \mathbb{N}_{2d}^n, \ |\psi_{-,\boldsymbol{\alpha}}| \leq \max\left\{\phi_\mathbf{0}, C\widetilde{R}^{2d}\right\}, \tag{62}
$$

where the term $C\widetilde{R}^{2d}$ is the same as in the proof of Theorem A.1. By Equations (44), (60), and (62), the feasible set of (50) is compact. As the cost is linear, there exists an optimal solution $(\boldsymbol{\psi}_+^{*(d)}, \boldsymbol{\psi}_-^{*(d)}, \boldsymbol{\phi}^{*(d)}) \in \mathbb{R}^{\mathbb{N}_{2d}^n} \times \mathbb{R}^{\mathbb{N}_{2d}^n} \times \mathbb{R}^{\mathbb{N}_{2d}^p}$.

*Part II: Convergence of the hierarchy.* Again it is similar to Part II in the proof of Theorem A.1. Extend $(\boldsymbol{\psi}_+^{*(d)}, \boldsymbol{\psi}_-^{*(d)}, \boldsymbol{\phi}^{*(d)})$ to infinite dimensional sequences, normalize via a scaling similar

---

[3]We can rewrite this as : $\left\langle \Sigma_+, L_{\boldsymbol{\psi}_+}(\boldsymbol{v}_d\boldsymbol{v}_d^\top)\right\rangle = L_{\boldsymbol{\psi}_+}(\boldsymbol{v}_d^\top \Sigma_+ \boldsymbol{v}_d) = L_{\boldsymbol{\psi}_+}(\sigma_+)$. Other expressions can be handled similarly.

to (47), apply Banach-Alaoglu theorem, and do the reverse scaling. Then deduce that there exists a subsequence $(d_k)_{k \in \mathbb{N}}$ and $(\psi_+^*, \psi_-^*, \phi^*)$ such that for all $(\boldsymbol{\alpha}, \boldsymbol{\beta}) \in \mathbb{N}^n \times \mathbb{N}^p$,

$$\lim_{k \to +\infty} \psi_{+,\boldsymbol{\alpha}}^{*(d_k)} = \psi_{+,\boldsymbol{\alpha}}^*, \lim_{k \to +\infty} \psi_{-,\boldsymbol{\alpha}}^{*(d_k)} = \psi_{-,\boldsymbol{\alpha}}^* \text{ and } \lim_{k \to +\infty} \phi_{\boldsymbol{\beta}}^{*(d_k)} = \phi_{\boldsymbol{\beta}}^*. \tag{63}$$

By closedness of PSD cone and linearity, the limiting sequence is feasible in (50) for every $d \geq d_{\min}$. In particular, this implies that $\mathbf{M}_d(\phi^*) \succeq 0$, and since $\mathcal{Q}_d(r)$ is Archimedean, by Putinar's Positivstellensatz, $\phi^*$ has a representing measure, $\phi^* \in \mathscr{P}(S_{\boldsymbol{\theta}})$.

Moreover, as $\boldsymbol{\mu}$ satisfies Assumption 2.1 and dominates $\psi_+^*$, $\psi_+^*$ satisfies Assumption 2.1 as well. Having $\mathbf{M}_d(\psi_+^*) \succeq 0$ for all $d \in \mathbb{N}$ then yields the existence of some representing measure $\psi_+^* \in \mathscr{P}(\mathbb{R}^n)$. Similarly, we can use Lemma A.1 to deduce that $\psi_-^*$ admits some representing measure $\psi_-^* \in \mathscr{P}(\mathbb{R}^n)$.

Finally, since (50) is a regularized relaxation of (16), we deduce that

$$\tau_{\varepsilon,R}^{\mathrm{TV}} \geq \tau_{d_k,\varepsilon,R}^{\mathrm{TV}} \xrightarrow{k \to +\infty} \psi_{+,\mathbf{0}}^* + \psi_{-,\mathbf{0}}^* + \varepsilon L_{\phi^*}(R). \tag{64}$$

Moreover, since $(\psi_+^*, \psi_-^*)$ is just one particular feasible solution, we necessarily have

$$\tau_{\varepsilon,R}^{\mathrm{TV}} \leq \psi_{+,\mathbf{0}}^* + \psi_{-,\mathbf{0}}^* + \varepsilon L_{\phi^*}(R). \tag{65}$$

Hence, $(\psi_+^*, \psi_-^*)$ is optimal in (16) for $\nu = \nu_{\phi^*}$. $\qquad\qquad\square$

Asymptotic convergence and existence of solutions at each level of the hierarchy, both guaranteed by Theorem A.2, make it now possible to derive the proof of Theorem 3.3.

*Proof of Theorem 3.3.* Existence follows from Theorem (A.2). To recover the best (in TV-regularized sense) $K-$atomic representation of $\mu$, proceed as in proof of Theorem 3.2. $\qquad\square$

# B  HANDLING HIGH-DIMENSIONAL DATA

## B.1  STANDARD DIMENSIONALITY REDUCTION

High-dimensional data present inherent challenges, as samples tend to be sparsely distributed in the ambient space, making structure difficult to detect. Dimensionality reduction techniques such as principal component analysis (PCA) and random projections have been shown to preserve both geometric and probabilistic structure under mild conditions (Dasgupta, 1999). In particular, random subspace embeddings ensure that spherical regions in the projected space retain approximately their expected number of sample points, thereby maintaining mixture separation. Within the Gaussian mixture framework, this observation motivates common simplifying assumptions, such as diagonal (not necessarily homoscedastic) covariance matrices, which facilitate modeling and computation without severely compromising representational power.

Building on these insights, we applied our method to the MNIST dataset. Specifically, we first reduced the dimensionality from $n = 784$ to $n = 2$ using PCA, thereby obtaining a low-dimensional representation that preserves the essential cluster structure. We then estimated the mixture order and its parameters via our approach, with the corresponding results presented in Figure 4.

## B.2  UNIVARIATE RELAXATIONS

Instead of directly approximating the measure $\mu$ supported on an $n$-dimensional space by a mixture $\nu_\phi \in \mathscr{P}(\mathbb{R}^n)$, we can consider its canonical *projections* (push-forward measures) $\pi_i \# \mu \in \mathscr{P}(\mathbb{R})$ and approximate them by mixtures of univariate distributions $\nu_{\phi^i} \in \mathscr{P}(\mathbb{R})$, where for each $i \in \{1, \dots, n\}$, $\phi^i \in \mathscr{P}(\pi_i(S_{\boldsymbol{\theta}}))$.

For each direction $i \in \{1, \dots, n\}$, we solve the following univariate problem:

$$\overline{\tau}_{\varepsilon,R}^{\mathrm{dist},i} := \inf_{\nu_{\phi^i} \in \mathscr{P}(\mathbb{R}), \, \phi^i \in \mathscr{P}(\pi_i(S_{\boldsymbol{\theta}}))} \left\{ \mathrm{dist}(\pi_i \# \mu, \nu_{\phi^i}) + \varepsilon R(\pi_i(\boldsymbol{\theta})) \right\}, \qquad \mathrm{dist} \in \{\mathrm{W}_2, \mathrm{TV}\}. \tag{66}$$

This yields $n$ separate univariate approximation problems, each of which can be efficiently addressed by SDP relaxations of order $d \geq d_{\min}$. Since the complexity of these relaxations grows with both

the ambient dimension and the relaxation degree, working in the univariate setting allows one to employ higher-order relaxations at manageable cost. As higher-order relaxations are typically more powerful, this substantially increases the likelihood of finite convergence and enables the extraction of high-quality estimates using Algorithm 1.

Let $\widehat{K}_i$ denote the estimated mixture order along the direction $i$. Although heuristic, it is natural to estimate the global mixture order by the *mode of the estimates* $\{\widehat{K}_i\}_{i=1}^n$. If there is no certainty about the most frequent value, this approach can still provide a finite candidate set of plausible values for $K$, somewhat analogous to confidence intervals in statistical inference.

**Remark B.1.** *In principle, an estimate $\{\widehat{\boldsymbol{\theta}}_j\}_{j=1}^{\widehat{K}}$ of the parameters of the multivariate mixture (our main object of interest) can be constructed a posteriori by stitching together the results obtained from the $n$ univariate cases. However, since each coordinate may yield a different number of candidate components, this reconstruction may result in a combinatorially large set of possible multivariate parameter estimates.*

As an illustration, we apply Algorithm 1 in order to solve SDP relaxations of problem (66). Results are presented in Table 1.

| | $\varepsilon = 10^{-5}$ | | | $\varepsilon = 0.1$ | |
|---|---|---|---|---|---|
| $\widehat{K}_i$ | Count | $\widehat{K}_i/n$ (%) | $\widehat{K}_i$ | Count | $\widehat{K}_i/n$ (%) |
| 1 | 161 | 20.54 | 1 | 163 | 20.79 |
| 2 | 60 | 7.65 | 2 | 140 | 17.86 |
| 3 | 80 | 10.20 | 3 | **452** | **57.65** |
| 4 | **461** | **58.80** | 4 | 8 | 1.02 |
| 5 | 22 | 2.81 | 5 | 21 | 2.68 |
| **Total** | **784** | **100.00** | **Total** | **784** | **100.00** |

Table 1: Distribution of univariate Gaussian mixture order estimates obtained by computing SDP relaxations $\overline{\tau}_{4,\varepsilon,R}^{\mathrm{W}_2,i}$ for $i \in \{1, \dots, n\}$, with two different values of the regularization parameter $\varepsilon$. MNIST data set with digits $\{0, 1, 2\}$ was used. We set $\mathtt{tol} = 10^{-6}$.

From Table 1, we can see that when $\varepsilon = 10^{-5}$, the modal outcome across coordinate-wise relaxations is $\widehat{K}_i = 4$ (58.8%), whereas for $\varepsilon = 0.1$ the dominant value shifts to $\widehat{K}_i = 3$ (57.7%). This behavior is consistent with the role of the regularization term: weaker penalization tends to overfit local irregularities, effectively splitting classes into additional components, while stronger penalization suppresses spurious complexity and is more likely to recover the true number of classes. Minor frequencies at $\widehat{K}_i \in \{1, 2, 5\}$ reflect directions with little discriminative signal or excess noise. In particular, many coordinates near the image borders contain only background pixels, so the distribution along those axes is essentially unimodal, giving rise to $\widehat{K}_i = 1$. Taken together, these results suggest that the candidate set of plausible mixture orders is $\{2, 3, 4\}$, with $\widehat{K} = 3$ favored under moderate regularization, in line with the ground-truth number of digit classes. Similar results were obtained by considering $\overline{\tau}_{4,\varepsilon,R}^{\mathrm{TV},i}$.

**Remark B.2.** *Another heuristic for estimating the mixture order in high dimensions is to project the data onto randomly chosen low-dimensional subspaces. For example, by considering $\overline{n} \ll n$ random one-dimensional projections, one can apply the same univariate relaxation strategy to obtain mixture order estimates. This may be advantageous compared to canonical (coordinate) projections, since random directions are more likely to cut through informative linear combinations of features rather than being restricted to individual coordinates. In settings such as images, many canonical directions correspond to near-constant background pixels and thus provide little signal, whereas random projections can capture structure spread across multiple coordinates, yielding more reliable order estimates.*

## C    EXPERIMENTAL SETUP

All experiments were performed on a computer with a 13th Gen Intel Core i7-13620H CPU @ 2.40 GHz, 10 cores, 16 logical processors, and 32GB of RAM. All the studied GMP instances were modeled using the Julia library TSSOS (Magron & Wang, 2021), and their corresponding Moment-SOS relaxations were solved with Mosek (Andersen & Andersen, 2000).

In all our experiments, input data was normalized to take values in the interval $[0, 1]$. EM algorithm was always run with 100 iterations and a convergence tolerance of $10^{-5}$.

**Experiments in Figure 2.**    As mentioned in the main text, Algorithm 1 identifies $\widehat{K} = 4$ components for either choice $\mathrm{dist} \in \{\mathrm{W}_2, \mathrm{TV}\}$. Namely, for the $\mathrm{W}_2$ distance, we obtain:

$$\widehat{\boldsymbol{m}}_{\mathrm{W}_2} = (0.1238, 0.8840, 0.5052, 0.5087),$$
$$\widehat{\boldsymbol{\sigma}}_{\mathrm{W}_2} = (0.1616, 0.1548, 0.4001, 0.1461), \tag{67}$$
$$\widehat{\boldsymbol{\phi}}_{\mathrm{W}_2} = (0.28045339, 0.26621474, 0.21218507, 0.24114680).$$

Moreover, for the TV distance, the components are given by:

$$\widehat{\boldsymbol{m}}_{\mathrm{TV}} = (0.1264, 0.5106, 0.8773, 0.5047),$$
$$\widehat{\boldsymbol{\sigma}}_{\mathrm{TV}} = (0.1509, 0.1441, 0.1593, 0.4053), \tag{68}$$
$$\widehat{\boldsymbol{\phi}}_{\mathrm{TV}} = (0.28645039, 0.23552885, 0.27091699, 0.20730377).$$

**Experiments in Figure 3.**    We have used the following description for the support of the mixing measure:

$$S_{\boldsymbol{m},\boldsymbol{\Sigma}} := \left\{ ((m_1, m_2), (\sigma_1, \sigma_2)) \in \mathbb{R}^2 \times \mathbb{R}^2 \mid (m_1, m_2) \in [0, 1]^2, (\sigma_1, \sigma_2) \in [0.05, 1]^2 \right\}. \tag{69}$$

Regularization parameter was set to $\varepsilon = 10^{-3}$, and the rank tolerance was $\mathtt{tol} = 10^{-2}$.

**Experiments in Figure 4.**    We have used the following description for the support of the mixing measure:

$$S_{\boldsymbol{m},\boldsymbol{\Sigma}} := \left\{ (\boldsymbol{m}, \boldsymbol{\sigma}) \in \mathbb{R}^2 \times \mathbb{R}^2 \mid (m_1, m_2) \in [0, 1]^2, (\sigma_1, \sigma_2) \in [5 \cdot 10^{-5}, 0.15]^2 \right\}. \tag{70}$$

As previously, we set $\varepsilon = 10^{-3}$ and $\mathtt{tol} = 10^{-2}$.

