# OpenReview forum: "Mixtures Closest To A Given Measure: A Semidefinite Programming Approach"
_ICLR.cc/2026/Conference — Submitted to ICLR 2026_

### Official Review · Reviewer_Skq7 · 2025-10-29

**Soundness:** 3
**Presentation:** 2
**Contribution:** 2
**Rating:** 2
**Confidence:** 3

**Summary:**

The paper applies a recently published approach to the approximation of W1 and TV distances between measures (Laserre 2024) to the specific case of approximating a given density by a finite mixture. The basic idea is to solve a semidefinite program based on a subset of  the moments of the finite mixture: by increasing the number of moments considered one can prove that eventually the recovered mixture will be optimal in terms of W1 or TV but of course this comes with a computational price since solving the SDP will be more expensive.

Some experimental results are shown on scalar and 2D problems. They show that initializing EM with the suggested method converges faster than EM with random initialization.

**Strengths:**

The paper addresses a difficult and foundational problem: even evaluating the TV between two given distributions is challenging so finding the optimal mixture using that criterion is a major challenge. The paper also brings the important advances of (Laserre 2024) to the attention of the ML community.

**Weaknesses:**

I am mostly concerned with the relevance to ICLR and the significance.

I believe the paper will be difficult to follow for most ICLR attendees. The derivation builds heavily on prior work (Laserre 2024a,b) and without reading those papers, it is hard to understand the contribution in the present paper.

The significance to a ML audience is limited by several factors. First, the computational cost is such that the method can only be applied to one dimensional or two-dimensional data. This is acknowledged by the authors and they suggest ways to allow scaling up in the appendix but in its current formulation the method is highly limited. Second, the benefit of using the method in an ML application is questionable. The experiments in the paper (using the method to initialize EM) show a decrease in the number of iterations required to achieve convergence but given the high computational cost of computing the initialization it is not clear that this is an attractive method for practitioners: I assume almost any practitioner would prefer to run EM for more iterations instead of solving an SDP.

Another issue of relevance is the reliance on TV and W1 metrics. For better or worse, most clustering algorithms use the likelihood objective which is equivalent to the KL metric.

**Questions:**

What is the motivation for using TV or W1 as initialization for clustering using EM and K-means (which are based on a KL divergence) ?

---

> ### Author Response · Authors · 2025-11-20
> **Detailed Response to Reviewer Skq7**
>
> We thank the Reviewer **Skq7** for providing a nice summary and acknowledging the strengths of our approach. However, we would like to point out that our approach is not meant to approximate a density (we do not assume that the unknown target measure has density). Additionally, we don't assume that the optimal mixture is finite. We also focus an $W_2$ distance, even though our approach can be extended to the $W_1$ case.
>
>
> **Weaknesses**
>
> *Sentences 1-6*:
>
> As the Reviewer **Skq7**  noted, the underlying problem is difficult and foundational, and bringing and further extending the recent advances of Lasserre (2024a,b) to the ML community is valuable despite their technical nature.
> As emphasized in our **General Reply**, we made a deliberate effort to keep the submission self-contained: only two core theorems are needed in the main text, the required background from Lasserre (2024a,b) is summarized directly, and all proofs are placed in the appendix.
> We believe the revised version improves readability while maintaining full rigor.
>
> Regarding significance and dimensionality, also discussed in the **General Reply**, we note that our approach is applicable beyond presented 1D–2D clustering experiments, which we nevertheless believe to effectively illustrate the potential of our approach in correctly estimating both mixture order and component parameters. In higher dimensions, one can still try computing low-order relaxations and extracting informative approximate mixtures, which can then be refined by gradient-based updates (as discussed in our answer to Question (7) by Reviewer **jxWS**) or by the projection-based methodology developed in Appendix B.2.
> Thanks to the feedback from Reviewer **jxWS**, we believe that the latter could be naturally combined with the theory behind sliced Wasserstein distances and provide a principled path toward scalability that we could not fully explore within the space constraints.
> Given its promising empirical performance (e.g., accurate order recovery on MNIST in Table 1), we estimate that this projection-based extensions could inspire exciting and original future works.
>
> *Sentences 7-8*:
>
> The goal of our method is not to replace or beat EM in large-scale practice but to provide a reliable, globally informed initialization.
> Repeated EM runs **do not offer global guarantees** and still **require the correct mixture order as input**, something that our method is demonstrated to efficiently recover.
> This makes the approach useful when the goal is to obtain a trustworthy mixture estimate rather than simply to minimize runtime, thereby justifying the computational cost. Finally, we are confident that approximate or projection-based variants of our approach could further reduce this cost in higher dimensions.
>
> *Sentence 9*:
>
> Finally, as this remark is similar to Reviewer's question, we provide the answer below.
>
> **Questions**
>
> Our choice of $W_2$ and TV is motivated by both modeling and computational considerations.
> Geometrically, unlike KL (which is an asymmetric information-gain measure whose minimization is only approximated via maximum likelihood), $W_2$ captures displacement structure and remains stable when data concentrate on low-dimensional manifolds, while TV measures mass mismatch.
> These properties make them well suited for producing coarse-scale, geometry-aware initial estimates that reliably guide EM and $k$-means toward good solutions, as observed in our experiments.
>
> Computationally, $W_2$ and TV admit polynomial, semi-algebraic dual formulations, allowing us to cast the corresponding mixture-approximation problem as a generalized moment problem and obtain **globally certified** approximations via the Moment-SOS hierarchy. In contrast, KL-based optimization landscapes are highly nonconvex, admit many poor stationary points, and offer no global guarantees.
> Their performance therefore depends critically on the quality of initialization and on providing the correct mixture order, which our method supplies.
>
> Recent work in (Bach, 2025), although not related to mixtures, shows that several $f$-divergences, including KL, can also be approximated via suitable SDP relaxations. Studying how this framework interacts with ours could eventually yield a global method for KL-based mixture estimation as well.
>
> Thus, our use of $W_2$/TV is not to replace KL-based clustering, but to provide robust, globally certified mixture-order and parameter estimates that can significantly improve the downstream performance of KL-driven methods.
>
> -- Bach, Francis. "Sum-of-squares relaxations for information theory and variational inference." *Foundations of Computational Mathematics* 25, no. 3 (2025): 865–903.

---

> ### Author Response · Authors · 2025-11-26
> **Response to Reviewer Skq7's additional feedback**
>
> We thank the Reviewer **Skq7** for the additional comment and for acknowledging the appeal of our work.
>
> Regarding the concern about using W2/TV to initialize KL-based optimization, we fully agree that the KL-optimal and W2/TV-optimal mixtures may differ, and we hope our work encourages further theoretical and applied research on elucidating these differences.
>
> Nevertheless, we believe that this difference is precisely why such initializations can still be helpful. KL (as optimized by EM) is known to be highly sensitive to initialization and can converge to poor local optima when components start far from meaningful regions of the data. In contrast, W2 and TV produce **geometry-aware approximations** that avoid degeneracies and are grounded in **global optimization principles**.
> Therefore, we believe it remains reasonable to use our estimates to provide a stable, globally certified starting point that places components in sensible locations before KL (EM) refinement.
>
> Moreover, our estimates do **not** have to be coupled **exclusively** with KL. As shown in Figure 2 (left column), the number of iterations of k-means (which is completely unrelated to KL) is reduced by **more than 30%**, which is substantial. Figure 3 supports this conclusion as well.
>
> In this sense, our method is powerful in (i) identifying the **correct number** of mixture components based on global optimization (a capability *unavailable* to classical heuristics), and (ii) providing high-quality initial parameter estimates that can serve as a **rigorous preprocessing step** for other scalable clustering algorithms.
>
> With this in mind, we believe that our contribution on **globally certified mixture *learning***, especially in its more accessible revised version (as acknowledged by Reviewer *jxWS*), is *meaningful for the ICLR community*, with the broader aim of motivating the development of more scalable global optimization tools that could benefit practitioners.
>
>
> We thank the Reviewer **Skq7** once again for engaging in the discussion and for the willingness to reconsider the revised version.

---

### Official Review · Reviewer_jxWS · 2025-10-30

**Soundness:** 2
**Presentation:** 1
**Contribution:** 3
**Rating:** 4
**Confidence:** 2

**Summary:**

This paper addresses the problem of approximating a target measure, known through finitely many of its moments, by a mixture from a parametric family (e.g., Gaussian, exponential, Poisson). The authors investigate optimality under two regularized moments
The authors propose a hierarchy of semidefinite relaxations for the corresponding optimization problem, with guarantees of asymptotic convergence and finite recovery under a rank condition. The work is mathematically rigorous and builds on the Lasserre hierarchy and related moment-SDP techniques.

Overall Assessment:
While the paper presents a mathematically elegant and theoretically sound framework, it is not presented in a manner accessible to the ICLR community, and the proposed approach does not scale beyond low-dimensional problems. The contribution may be more suitable for a venue focusing on mathematical optimization or computational statistics, rather than a machine learning conference that emphasizes scalable algorithms and empirical evaluation.

**Strengths:**

- The paper has a very clearly defined objective and proposes a mathematically precise solution via optimality conditions.
- Gaussian mixture continues to be highly relevant for several tasks.
- Research into mixtures is a very relevant topic
- I think that the interplay between measure optimality with respect to different divergences is very interesting
- Overall, the paper has a mathematically very interesting foundation,

In conclusion, this is a very interesting paper from the point of view of mathematical optimization and computational statistics.

**Weaknesses:**

*Main Text*:

- The paper is very closely related to (Lasserre, 2009, 2015, 2024 a, 2024b) and not very intuitive for the uninitiated, e.g., lines 199-202. While I think that this is certainly a very interesting line of research, understanding this paper in detail requires an in-depth understanding of previous works.

- Structure: Choosing Wasserstein and TV seems somewhat arbitrary at first glance. While it makes sense given the existing literature, it would make sense to discuss potential extensions to MMD, sliced Wasserstein and other divergences.

- Mathematical statements in the main text are not formulated very clearly, e.g., Thm 1. While the supplementary provides an exact formulation, the formulations in the main text seem rather vague.

- Line 187: “ The optimization problem in (7) is formulated over
probability measures, making it infinite-dimensional, and thus intractable in its original form.” Note that this might be somewhat misleading since parametric mixtures and empirical measures are not infinite-dimensional. Maybe the conditions should be specified.

- lines 255-257: “However, the sample size $N$ is often large enough, and therefore, by the law of large numbers, the difference between empirical and true population moments is negligible.” I think that the joint estimation of all moments depends on the ambient dimension $d$. Therefore, the phrase in the paper might be misleading, i.e., $N$ would have to be *very* large in high dimensions. Please comment on the potential impact of the curse of dimensionality.

*Numerical Evaluation*:

- The paper ‘Sliced Wasserstein Distance for Learning Gaussian Mixture Models’ (Kolouri et a., 2017) proposes another method to fit Gaussian mixtures with a (sliced) Wasserstein distance. While there are key differences, I think that this is a relevant reference and that there should be a numerical comparison between this method and the proposed one. Similarly, the cited paper ‘Learning Gaussian Mixtures using the Wasserstein-Fisher-Rao Gradient Flow’ (Yan et al., 2024) learns Gaussian mixtures using the Wasserstein-Fisher-Rao geometry, and I would have appreciated a numerical comparison.

- Figure 1: This is an interesting visualization, but then again, many methods have been proposed to curb the number of mixture components. Moreover, the regularization term works as a kind of prior on the component number. Thus, a comparison with Bayesian mixture techniques (Richardson et al., 1997; Nobile, 2005; Miller et al., 2018) would have made more sense.

- Figure 2: Please note that k-means is a highly efficient and fast algorithm and that there exist many fast and effective initialization heuristics beyond ‘random’ ones. Moreover, there is no clear advantage of ‘TV’-initialization in general and no clear advantage of ‘Wasserstein’-initialization for large ‘mixture indices’. Moreover, the runtime of initialization+K-means should be reported since this is the value that is actually interesting in practice.
- Figure 3: This experiment gives no additional insight beyond the ones presented in Figure 2. Projecting MNIST onto 2 dimensions is not really a high-dimensional experiment.
- Gaussian mixtures allow for very nice visualizations in 2D. Therefore, I would have appreciated a density heatmap or a similar plot that showcases the behaviour of the Wasserstein-optimal vs. the TV-optimal vs the KL-optimal (obtained via EM) mixtures for a fixed number of components. Given that the paper focuses on such settings, readers would have gained a more intuitive understanding.


*Conclusion*:
Despite the interesting theoretical background, the paper suffers from a lack of realistic high-dimensional experiments, comparative/ablation studies, and visual intuition. Moreover, the contents of this paper are not presented in a very accessible manner and are hard to follow for readers who are not familiar with (Lasserre, 2009, 2015, 2024a, 2024b). I would recommend that the authors add more visual studies and investigate the adaptation of their optimality formulations to higher dimensions. Personally, I would find a thorough investigation of the interplay between KL, Wasserstein, and TV optimality for 2d mixtures highly interesting.

**Questions:**

- I would have been interested in the impact of optimizing a given divergence on another divergence. What is the interplay of TV and Wasserstein? What is the interplay of KL (as used by EM) and Wasserstein/KL beyond simple 1d uniform distributions?
If you minimize the sliced Wasserstein using the algorithm of (Kolouri et al., 2017), what is the impact of the Wasserstein (and vice versa)?

- Can you quantify the number of constraints with respect to the dimension?

- There exists a paper on reducing mixture components using Wasserstein distance (Assa et al., 2018). Given the goal of reducing the components of a target mixture, how could one compare their reduction outcome with your method?

- Beyond CV and BIC, how does your component number estimation compare to estimation via Bayesian methods, see (Richardson et al., 1997; Nobile, 2005; Miller et al., 2018)? I feel as if this is closer to the presented work since a regularization term/component number prior is used.

- What is the impact of varying the regularization? I assume that the smoothness of the estimated mixtures in Figure 1 is at least partly the result of the regularization and not of the Wasserstein/TV optimality.

- A whole line of research, e.g., (Delon et al, 2019; Dusson et al., 2023) and more recently (Nguyen et al.; Bonet et al.; Piening et al., all 2025), is devoted to developing Wasserstein-type metrics between mixtures. While they are not meant to fit mixtures, they are meant to compare between a target and a reference mixture. Did you consider metrics of this kind for your evaluation? Do you see any relevance of the studied `Mixture Wasserstein'/`Wasserstein-over-Wasserstein'-type metrics to your work?

-  There are papers (Kolouri et al, 2017; Geppert et al., 2021) investigating mixture estimation via gradient descent. Did you consider such gradient-based solvers to `approximately’ solve the optimization problem to enable approximate solutions to the proposed optimization problem in higher dimensions?

- The proposed discrepancy-regularization reminds me of research on Wasserstein gradient flow for regularized f-divergences, see (Stein et al, 2025). While this work is not directly relevant, do you think that you could extend some of your results to general f-divergences?

---

> ### Author Response · Authors · 2025-11-20
> **Detailed Response to Reviewer jxWS - I**
>
> We thank the Reviewer **jxWS** for providing a nice summary and acknowledging the strengths of our approach.
>
> **Weaknesses**
>
> *Main text*
> -  As stated in our **General Reply**, we propose a hierarchy specifically designed for mixture approximation under **both $W_2$ and TV distance**. It (i) introduces a **regularization term** that stabilizes the SDP relaxations, (ii) extends the formulation to **multivariate and non-Gaussian mixtures**, and (iii) provides an **explicit procedure** to recover both the mixture order and component parameters. While the framework is necessarily technical, we have made significant efforts to summarize all essential background so that it can be followed independently of prior works, while also **extending and operationalizing** this theory in a new, practically relevant setting. We hope the revised version highlights these aspects more clearly.
> - We appreciate the Reviewer’s suggestion and emphasize that the choice of $W_2$ and TV is not arbitrary but grounded in both theoretical and modeling considerations. These two metrics capture complementary aspects of distributional discrepancy: $W_2$ quantifies geometric displacement between measures, while TV reflects mass mismatch. Both have well-established statistical interpretations and are directly relevant in machine learning contexts involving transport or robustness. Moreover, when one of the distributions is assumed to be a mixture, the corresponding distance minimization can be formulated as a Generalized Moment Problem (GMP), enabling rigorous Moment-SOS relaxations with global convergence guarantees. Regarding other divergences, our approach would naturally *extend to any distance admitting a semi-algebraic representation*. This includes, for instance, MMD with polynomial kernels. We have mentioned these connections in the revised version (lines 533-536).
> - We appreciate the remark. In the revised version, we have reformulated the relevant mathematical statements to remove potential ambiguities, while keeping the exposition as clear and minimally technical as possible.
> - We thank the Reviewer for this observation. However, even when replacing $\mu$ by its empirical counterpart $\mu^N$, the mixing measure is not assumed to be discrete but rather an arbitrary probability measure supported on a parameter set that is compact and semi-algebraic rather than finite. Hence, the problem in (7) remains infinite-dimensional and intractable in its original form. As we show, it nevertheless admits exact moment reformulations for the considered distances, which in turn yield finite-dimensional SDP relaxations. We have emphasized this in our revised version (lines 186-189).
> - We agree that the ambient dimension $n$ influences the sample size $N$ required for accurate empirical moment estimation. In particular, by Corollary 6.3.2 in (Lasserre et al., 2022), for a relaxation of order $d$, it suffices that $N \ge \max\{C, \binom{n+d}{d}\}$, for some moment-dependent constant $C>0$, to guarantee that empirical and theoretical moments are arbitrarily close. This condition is satisfied in all our experiments. Importantly, our framework relies only on moments up to (preferably small) fixed order $d$, rather than on all high-order moments, which mitigates the curse of dimensionality. The same reference also provides guidelines on choosing relaxation order, as a function of sample size, in order to maintain high approximation quality. We have clarified this point in the revised version (lines 294-298).
>
> -- Lasserre, J.B., Pauwels, E. and Putinar, M. (2022) The Christoffel–Darboux Kernel for Data Analysis. Cambridge University Press (Cambridge Monographs on Applied and Computational Mathematics).

---

> ### Author Response · Authors · 2025-11-20
> **Detailed Response to Reviewer jxWS - II**
>
> *Numerical Evaluation*
>
> - We thank the Reviewer for pointing out this reference. In (Kolouri et al. (2017)), the authors fit GMMs by minimizing a *sliced* Wasserstein objective (an average of 1D $W_p$ distances over random projections) via sample-based, gradient-driven optimization with user-specified hyperparameters (e.g., number of slices, step size). The mixture order $K$ is also assumed known a priori. While computationally efficient and suitable for high-dimensional data, this method remains heuristic and provides only local convergence. In contrast, our framework tackles a different regime and objective: we formulate *regularized distance minimization (for both TV and $W_2$)* to an arbitrary target measure, known only through its **moments**, as a Generalized Moment Problem and solve it using Moment-SOS relaxations that enjoy **global convergence guarantees**, allow **component extraction**, and extend naturally to **non-Gaussian** mixture families. Our approach can also recover the mixture order through rank conditions, **without** assuming $K$ a priori.
> Interestingly, our projection-based variant presented in *Appendix B.2 (univariate approximations)* shares conceptual similarities with the sliced Wasserstein approach. Given the strong empirical performance of our univariate relaxations in correctly estimating mixture order, we believe that a relevant future research topic is a thorough study of such projection-based adaptations within our framework that could substantially improve scalability and extend this ''slicing principle'' to other divergence measures. **Lastly**,
> we note that the setting in Yan et al. (2024) is quite different from ours. They propose a continuous-time, gradient-flow method for learning Gaussian mixtures under the Wasserstein-Fisher-Rao geometry. Their approach relies on parametric updates of mixture components through partial differential equations and requires access to samples and differentiable likelihoods. The method also needs to be initialized with a mixture of correct order (often not known a priori) and applies only to isotropic Gaussian mixtures, whereas Gaussian mixtures within our framework can have arbitrary covariance structures.
> - We would like to clarify that the purpose of Figure 1 (now Fig. 2) was not to demonstrate the ability of our method to curb the number of mixture components, but rather to illustrate the nature (or quality) of the recovered mixtures (TV/$W_2$-based geometric constraints seem to favor smoothness and avoid overfitting). As stated in the caption, we set $\varepsilon = 0$, so no regularization/prior was applied. Interestingly, even in this unregularized setting, our method showed a natural tendency not to overfit, unlike the two widely used competing approaches included for comparison.
> - We agree that both $k$-means and EM are efficient algorithms; however, our approach is not meant to compete with them directly, but rather to provide high-quality warm-start initializations. Importantly, before initialization, our method also offers a **certifiable estimate of the mixture order** through a rank condition, which is something we are not aware of in existing approaches. Running even the best initialization heuristics with an incorrectly estimated number of components can lead to poor final models. As shown in Figure 2 and discussed in the corresponding paragraph, our initializations yielded the correct mixture order and parameters, reducing the number of iterations by about 90% for $k$-means and 67% for EM on average. We also observed that TV-based initializations performed better when combined with EM, while $W_2$-based ones favored $k$-means. Since we used low relaxation orders, the corresponding runtime remained moderate; we have included runtime statistics the revised version (lines 468-470).
> - While the real dimension in both experiments might be the same, we strongly believe that Figure 3 **does provide** additional insight.
> Namely, Figure 2 studies the idealized setting where data are truly Gaussian, and our method correctly recovers both the mixture order and accurate parameter estimates.
> Figure 3 serves a different purpose: MNIST is **not** Gaussian, yet we show that projecting onto a low-dimensional feature space and fitting the *best* $W_2$/TV-regularized Gaussian mixtures still yields correct mixture order and parameter estimates. We note that such projection/reduction-based modeling is common in practical machine learning applications when only a few informative features are retained.
> Hence, Figure 3 can be understood as illustrating robustness of the method to model misspecification, **complementing and reinforcing**, rather than duplicating the conclusions of Figure 2.

---

> ### Author Response · Authors · 2025-11-20
> **Detailed Response to Reviewer jxWS - III**
>
> - We would like to clarify that, since BIC is KL/likelihood-based, our goal in Figure 1 was precisely to illustrate the recovered mixture supports and the qualitative differences between $W_2$/TV/KL-based approximations. The $W_2$ and TV distances capture geometric discrepancies between measures, whereas KL is purely information-theoretic; locally maximizing likelihood can therefore induce overfitting, which our approach seems to avoid naturally. In the subsequent experiments, EM-based mixtures were initialized with parameter estimates from our method to help the algorithm escape poor local minima and achieve better convergence. Finally, as suggested, we agree that systematically studying which distance is the most suitable for which data types and parametric families represents an interesting future direction.
>
> *Conclusion:*
>
> We appreciate these constructive suggestions. Our primary focus in this paper was to establish a rigorous and general framework for mixture approximation under $W_2$ and TV distances, with certifiable convergence guarantees and practical extraction procedures. While our experiments are conducted on benchmark instances of moderate size, they already clearly demonstrate the practical effectiveness of the proposed regularized hierarchy, and tailored implementations are likely to yield even better performance.
> Thus, our method is not inherently limited to such settings: low-order relaxations remain usable
> in higher dimensions, and projection-based variants (Appendix B.2) offer a
> clear path toward further scalability. We also agree that studying the interplay between KL/$W_2$/TV-optimal mixtures, e.g., as a function of separability or covariance structure, would represent a valuable contribution. Given the current paper length, we believe that detailed studies of such projections and interplays constitute a promising direction for future work.

---

> ### Author Response · Authors · 2025-11-20
> **Detailed Response to Reviewer jxWS - IV**
>
> **Questions**
> - As discussed above, we believe that our univariate projections approach from Appendix B.2 can be interpreted as a generalization of the sliced Wasserstein distance to the setting where the optimal mixture order is unknown. In our MNIST experiment, we used canonical projections along coordinate axes for clarity of illustration. We believe that coupling our framework with more efficient and principled projection strategies, e.g., such as small number of well-chosen random projections, as commonly done in the sliced Wasserstein literature, and linking univariate estimates back to the ambient-space representation could substantially improve scalability. As noted after Table 1, we did not observe significant differences between the TV- and $W_2$-based formulations in this projection ("slicing") experiment.
> - In (12) (now (35)), the SDP constraints are decoupled between $(\boldsymbol{x},\boldsymbol{y}) \in \mathbb{R}^{2n}$ and the parameter space $\boldsymbol{\theta} \in \mathbb{R}^p$.
> The main moment block $\mathbf{M}_d(\boldsymbol{\lambda}) \succeq 0$,  has size $s(n,d) = \binom{n+d}{d}$,
> and each localizing block
>
> $\mathbf{M}_{d-d_j}(r_j \boldsymbol{\phi}) \succeq 0$ has size $s(p,d-d_j) = \binom{p+d-d_j}{d-d_j}$.
>
> Linear equalities (12b) contribute about $2 s(n,2d) = 2 \binom{n+2d}{2d}$ constraints.  Hence, for fixed $d$, the number of constraints grows polynomially with $(n,p)$,  dominated by the larger SDP block of size $\max[s(2n,d),\{\max_j s(p,d-d_j)\}]$.
> Analogous scaling holds for the regularized TV formulation.
> - We thank the Reviewer for pointing out this related work.
> The setting in Assa and Plataniotis (2018) is, however, fundamentally different from ours.
> Their method assumes that the full Gaussian mixture model is already **given**, including all component weights, means, and covariances, and the goal is to **reduce** this mixture by merging or pruning components so as to obtain a smaller mixture that remains close to the original one under a Wasserstein-based cost.
> In other words, they solve a mixture compression problem over known parametric mixtures.
> In contrast, our framework **does not assume** that initial target distribution is a mixture, nor that any mixture parameters (or even the mixture order) are known.
> We tackle the more general problem of approximating an arbitrary probability measure given only finitely many of its moments. Therefore, we believe that the two approaches address different problem regimes, have different problem data, and are not directly comparable in practice.
> - We thank the reviewer for highlighting these classical Bayesian approaches.
> While they are indeed designed to estimate the number of mixture components, their setting is fundamentally different from ours.
> Bayesian methods such as Richardson-Green (1997), Nobile (2005), and Miller-Harrison (2018) rely on full likelihood-based inference, require access to samples and evaluable component densities, and explore the posterior over $(K,\theta)$ via MCMC (e.g., reversible-jump or allocation samplers).
> In contrast, our problem is a *moment-based distance minimization* task: the unknown target distribution is **not assumed to have a density**, may not admit an exact mixture representation, and is available only through finitely many empirical or theoretical moments.
> In this regime, Bayesian model-selection techniques are not applicable, whereas our rank-based identification is.

---

> ### Author Response · Authors · 2025-11-20
> **Detailed Response to Reviewer jxWS - V**
>
> - We clarify that no regularization was used in Figure 1 ($\varepsilon=0$),
> so the smoothness observed there comes purely from the geometry induced by $W_2$/TV optimality conditions.
> As observed in Appendix B.2, increasing the regularization parameter $\varepsilon$ mainly accelerates finite convergence of the hierarchy by promoting both SDP stability and sparsity in the mixing measure. We believe that even a relatively weak regularization is sufficient to achieve this behavior and maintain the interpretability of optimal solutions.
> - We did not use these mixture Wasserstein metrics in our evaluation, since, as the Reviewer notes, they require both the target and the reference distributions to be given explicitly as Gaussian mixtures with known weights, means, and covariances.
> However, there might be an indirect relevance to our setting in the following sense: once a mixture has been recovered by our method, such distances could be used to compare it with mixtures obtained by alternative procedures, but they do not assist in estimating the mixture itself from moment information.
> Given that $W_2$/TV/KL optimality may lead to different recovered mixtures, using these mixture-mixture metrics in future work could indeed help elucidate the differences between them and shed light on the interplay between these divergences.
> - Given the global-optimality-driven nature of our method, we did not directly investigate local gradient-descent approaches.
> In high-dimensional settings, however, only low-order relaxations may be computationally feasible; while these may not certify global optimality, they still yield approximate mixtures via a generalization of Algorithm 2.
> Such approximate solutions could then be refined using gradient-based methods, eventually providing a practical alternative when increasing the relaxation order is not possible.
> - We believe that extending the framework to other $f$-divergences is an interesting direction.
> Our methodology can indeed accommodate any divergence whose dual or variational representation leads to a GMP with polynomial or semi-algebraic data. We believe that recent work (Bach, 2025) might provide relevant insights for these extensions.
>
> -- Bach, Francis. "Sum-of-squares relaxations for information theory and variational inference." Foundations of Computational Mathematics 25, no. 3 (2025): 865-903.

---

> > ### Comment · Reviewer_jxWS · 2025-11-24
> > **Thanks**
> >
> > Thank you for the very detailed answers and for revising the manuscript! Based on the rebuttal, I raised my score.
> >
> > While the new version is more accessible, I still believe there is room for improvement in this area. Given the limited real-world applicability of the proposed algorithm, I would further argue that it is mostly interesting for low-dimensional settings and from a theoretical point of view. Therefore, the paper would have benefited from a stronger focus on informative and well-visualized toy experiments.
> >
> > Nevertheless, the paper is interesting from a theoretical viewpoint.

---

> ### Author Response · Authors · 2025-11-26
> **Response to Reviewer jxWS's additional feedback**
>
> We thank Reviewer **jxWS** for taking the time to reconsider their score and for the positive feedback regarding the revised version.
>
> As acknowledged in our General Comment, our primary goal in this work is to establish a rigorous, globally convergent framework for mixture approximation under W2/TV—problems that are fundamentally challenging even in low dimensions.
>
> Our experiments indicate that KL, W2, and TV may emphasize different geometric or statistical aspects of mixture estimation. Understanding these differences in a principled way for different (even low-dimensional) mixture families (Gaussian, Poisson, exponential, etc) is a compelling avenue for future research, and our work provides a starting point for such analyses.
>
> While the current hierarchy is most practical in low to moderate dimensions, we emphasize that the **framework itself is general**, and that the projection-based variants discussed in Appendix B.2 offer a concrete path toward scalable extensions. Our hope is that this work will stimulate further research, both theoretical and algorithmic, on **globally certified mixture learning**, an area that is still in its early stages.
>
> We sincerely appreciate Reviewer **jxWS**’s constructive engagement, the acknowledgment of the theoretical interest of our contribution, and the improved evaluation.

---

### Official Review · Reviewer_AYHy · 2025-10-31

**Soundness:** 2
**Presentation:** 2
**Contribution:** 2
**Rating:** 4
**Confidence:** 2

**Summary:**

The paper studies the problem of approximating a target probability distribution (known through its moments) using a mixture of distributions. The goal of the proposed method to identify the mixture that is closest to the target under either the 2-Wasserstein or Total Variation (TV) distance. The paper assumes that the parameters of the mixture components lie within a compact semi-algebraic set. Combined with the assumption of polynomial moments, this formulation allows the problem to be converted into a Generalized Moment Problem (GMP). Furthermore, when the distance between the target and the learned distribution is measured using the 2-Wasserstein or TV metric, the relaxation of this problem can be expressed as a Semidefinite Programming (SDP) problem. The paper also presents experiments on synthetic data and the MNIST dataset to evaluate the practical effectiveness of the proposed approach.

**Strengths:**

The paper addresses an important problem of learning with mixtures of distributions, where each component is constrained to lie within a compact semi-algebraic set. The formulation is theoretically elegant. It is nice to see that the authors also evaluate the practical effectiveness of their proposed algorithm.

**Weaknesses:**

- I am not entirely sure about the technical novelty of this work compared to prior studies, particularly Lasserre (2024a) and Lasserre (2024b). The SDP relaxation of the Wasserstein distance appears closely related to Lasserre (2024a), while the relaxation of the TV distance seems inspired by Lasserre (2024b). One of the main contributions seems to lie in formulating the problem itself—such as defining each mixture component within a compact semi-algebraic set. As a result, the novelty of the proposed technique and/or results appears somewhat limited. Given that the experimental performance of the algorithm is also not particularly strong, the overall technical contribution remains unclear.

- The paper is also difficult to follow in several sections, especially for a venue like ICLR. Readers may find Sections 2 and 3 overly technical. It may be beneficial to present these sections in a slightly informal manner, while moving the formal statements and results to the appendix.

Minor:

- The following paper [1] also considers the problem of estimating the number of components in Gaussian mixtures, so it is somewhat related to this work considering this work also considers the problem when the mixing distribution is unknown.

[1] Robust Model Selection and Nearly-Proper Learning for GMMs. Liu et al. 2022

**Questions:**

See weakness

---

> ### Author Response · Authors · 2025-11-20
> **Detailed Response to Reviewer AYHy**
>
> We thank the Reviewer **AYHy** for providing a nice summary and acknowledging the strengths of our approach.
>
> **Weaknesses**
> - We invite the Reviewer to consult our **General Reply** (**Novelty**), where clarifications about novelty are provided. Distinct features of our paper have been emphasized in the revised version (e.g., contribution paragraph, lines 319-320, etc) as well.
> - As explained in our **General Reply**, our goal was to keep the paper accessible yet self-contained. The main text therefore presents only the two key theorems that underpin Algorithm 1, while technical lemmas, existence results, and proofs have been moved to the Appendix. We have also made additional efforts to make the paper more *ICLR-friendly*: for example, by writing our SDP relaxations (12) and (17) in a very generic and easy to understand form (8) (explicit details were moved to Appendix), or by providing clear pipeline overview (see Figure 1). We invite the reviewer to consult the revised version.
>
> *Minor*
> - We thank the Reviewer for his reference to Liu et al. (2022), which addresses the problem of robust model selection, where the unknown measure is **supposed to be a univariate Gaussian mixture**. Their goal is to determine the smallest number of components $K$ needed so that an empirical distribution is $\varepsilon$-close in TV distance to a true GMM, providing sample-complexity and robustness guarantees. In contrast, our work formulates a fundamentally different problem: given any (possibly continuous or high-dimensional) measure, we compute the mixture that minimizes a chosen distance ($W_2$ or TV) to it. The result is a convex, globally convergent hierarchy that extends to multivariate mixtures and arbitrary parametric families, not just Gaussians. We believe that the two approaches are thus complementary in scope and methodology.

---

### Official Review · Reviewer_8vsn · 2025-10-31

**Soundness:** 2
**Presentation:** 3
**Contribution:** 2
**Rating:** 4
**Confidence:** 2

**Summary:**

The authors develop semidefinite programming (SDP) relaxations for computing the W2 and TV distances between two probability measures within a Moment-SOS framework (Lasserre, 2009), which has suitable regularization to mitigate numerical instabilities and promote sparse support as well. They tested the proposed methods on a range of unsupervised clustering tasks on the underlying
parametric mixture family, which produces mixture estimates that can significantly improve the performance of standard clustering algorithms.

**Strengths:**

The paper addresses approximating (via either W2 or TV distance minimization) an arbitrary given measure by mixtures from specific parametric families by introduceing two regularized Moment-SOS-type hierarchies of SDP relaxations.

As for the proposed method, the authors establish comprehensive theoretical results of the convergence guarantees.

Empirical results confirm the method’s performance gains among clustering tasks under GMMs.

**Weaknesses:**

1. There is ambiguity or confusion in the statement of theorems. In Theorem 3.2 and Theorem 3.3, The phrase “μ can be best approximated by …” is ambiguous and needs clarification. Please state explicitly the admissible set against which the optimum is taken.

\
2. In practical applications, Algorithm 2 involves Cholesky factorization and Schur decomposition, which can be computationally expensive and may limit its scalability. Therefore, it might not be as efficient as lightweight alternatives such as k-means++. Moreover, several recent studies have proposed preprocessing techniques for clustering that provide strong initial parameter estimates, such as semidefinite programming (SDP)–based methods by Chen and Yang (2021) and Zhuang et al. (2024). Have you compared your algorithm with these approaches to evaluate computational efficiency and initialization performance?

\
3. The current applications are limited to clustering tasks under Gaussian Mixture Models (GMMs). Could the authors discuss how their proposed methods performed in experiments when applied to other problems beyond GMM clustering?



\
[1] Xiaohui Chen and Yun Yang. Cutoff for exact recovery of gaussian mixture models. IEEE Transactions on Information Theory, 67(6):4223–4238, 2021.
[2] Yubo Zhuang, Xiaohui Chen, Yun Yang, and Richard Y. Zhang. Statistically optimal k-means clustering via nonnegative low-rank semidefinite programming, 2024. URL https://arxiv.org/abs/2305.18436.

**Questions:**

See weakness

---

> ### Author Response · Authors · 2025-11-20
> **Detailed Response to Reviewer 8vsn**
>
> We thank the Reviewer **8vsn** for providing a nice summary and acknowledging the strengths of our approach.
>
> **Weaknesses**
>
> 1. We appreciate the remark. To avoid introducing supplementary notation, feasible sets are always defined by constraint numbers. For example, for problem (10), it is thus given by equation (10b). We have provided improved theorem statements in the revised version.
>
> 2. The dominant computational cost lies in solving the SDP relaxations, not in the extraction step of Algorithm 2. Both Cholesky and Schur decompositions have efficient parallelizable implementations, making this step relatively inexpensive and numerically stable, particularly when the number of mixture components is moderate, as is typical in practice. We agree with the Reviewer that *k*-means++ is lightweight and efficient; however, it (i) remains heuristic and (ii) requires the number of clusters $K$ to be specified in advance. In contrast, **deterministic** points recovered by Algorithm 2 correspond to **globally optimal mixtures** identified through our distance-minimization framework.
> Finally, we thank the Reviewer for the suggested references. While related in spirit ((Zhuang et al. (2024)) does not consider mixtures estimation), our framework fundamentally differs: it is **distance-minimization-based**, does **not require $K$ to be known a priori**, and makes **minimal assumptions** on the underlying distributions (for instance, in the Gaussian case, we do not assume equal or known variances).
>
> 3. We would like to emphasize that our framework is not restricted to Gaussian mixtures. It applies to any family of distributions with polynomial moments satisfying Carleman condition. This includes, for example, Poisson, exponential, and other common exponential-family distributions. In fact, the proposed formulation and convergence guarantees remain valid in all such cases, and extending the empirical evaluation to these families is a natural direction for future work. Since our hierarchies provide a principled tool for computing globally optimal mixture approximations under $W_2$ or TV, we believe it could be useful, for example, for the following applications beyond clustering:
>
>   (i) **Distributionally robust optimization**, where it can enable constructing tractable **mixture-based** ambiguity sets within a certified $W_2$/TV radius;
>   (ii) **Out-of-distribution detection and anomaly scoring**, where estimating the closest mixture model under $W_2$ or TV provides a principled surrogate for the nominal data distribution. Indeed, lower bounds from our hierarchy can be understood as estimating quantitatively how far is the data from being generated by a mixture;
>   (iii) **Mixture reduction and probabilistic compression**, where our framework, with suitably chosen regularizations, can provide globally optimal reduced mixtures (i.e., models with lower complexity) with moment-matching guarantees, unlike heuristic merging methods.
>
> Adapting our proposed framework for these applications could be a topic for future work.

---

### Author Response · Authors · 2025-11-20
**General Reply - Part I**

Dear Reviewers,

First of all, we thank you all for your time, valuable feedback and constructive comments. It has been very useful for enhancing the overall quality of our paper. (All modifications in the revised manuscript are highlighted in **blue**)

**Strengths:**

We greatly appreciate the Reviewers’ recognition of the key strengths of our work.

Reviewers generally recognized the importance and relevance of advancing computational frameworks for approximating probability measures by parametric mixtures.

Notably, Reviewers **8vsn**, **AYHy**, and **jxWS** praised the work for its *comprehensive theoretical guarantees* and its *mathematically elegant* and *theoretically sound framework*. In addition, Reviewer **Skq7** appreciated our efforts in *attracting attention of ML community* to this important problem. Finally, Reviewer **8vsn** acknowledged our *method’s performance gains among clustering tasks under GMMs*.

**Limitations:**

We would also like to express our gratitude to all the Reviewers for their meaningful critiques and suggestions. Below, we address some common points and concerns that were raised.

- **Relevance to ICLR**: Reviewer **Skq7** primarily raised concerns about the relevance of our submission to the ICLR community. We would like, however, to note that ICLR, and other major ML venues, have a strong tradition of welcoming theoretically grounded works that bridge machine learning and optimization. This is precisely the goal of our paper. The proposed framework provides **global convergence guarantees**, which are rare in the ML literature where most algorithms are inherently local. We believe that exposing the ICLR community to such globally convergent optimization techniques can inspire new ways to tackle hard nonconvex problems more efficiently, thereby enabling a broader range of applications. In this sense, our work contributes to strengthening the dialogue between optimization theory and modern machine learning, a direction that has already proven highly fruitful in recent ICLR research.
- **Technical side:** Reviewers **AYHy**, **jxWS** and **Skq7** expressed their concerns regarding the paper's accessibility. Our goal was to maintain mathematical rigor while keeping the paper self-contained, so that readers need not rely on prior works such as Lasserre (2009–2024) to follow the main arguments. Sections 2–3 include formal statements only where necessary for clarity and completeness, while the core intuition is already conveyed through summary/interpretation paragraphs (e.g., in revised version: lines 206-215, complete introduction to Section 3, lines 269-272, lines 284-289, lines 341-348) and illustrations (e.g., in revised version, Figure 1 and Figure 2). We believe that our revised version further enhances readability by reorganizing these sections, providing clearer explanations of the main ideas, and moving some of the more technical details to the appendix, while the overall presentation already clearly conveys the key concepts and contributions, making it accessible to readers with limited prior familiarity.
- **Novelty**: Reviewer **AYHy** expressed some difficulty in identifying the technical novelties of our work. As correctly noted by reviewers **jxWS** and **Skq7**, the paper indeed builds upon the theoretical foundations of Lasserre (2024a,b), but it introduces several key advances beyond them.
**First**, our framework is strictly more general, as it incorporates a regularization (generalization) term that improves the numerical stability of the resulting SDPs and enhances modeling flexibility depending on the application and data at hand, which we believe to be a non-trivial extension.
**Second**, our results apply to general multivariate mixtures with polynomial moments, not only to Gaussian ones. Notably, mixtures were not treated at all in Lasserre (2024b). In contrast, our Lemma A.1, which can be viewed as an *independent theoretical contribution*, provides explicit analysis for Gaussian, Poisson, and exponential mixtures and could naturally extend to other parametric families.
**Third**, we propose an explicit algorithmic formulation that inherits the theoretical convergence guarantees while remaining practically relevant for mixture-learning and clustering tasks. We believe this combination of theoretical generalization, mixture-specific analysis, and implementable optimization framework represents a substantial step forward and may inspire further developments with even broader empirical reach.
**Finally**, we believe that (i) the polynomial property of moments satisfied by several important parametric families (see Assumption 2.2), and (ii) the fact that such mixtures satisfy Carleman's condition (Lemma A1), are not widely known/exploited in the ML and optimization literature. One by-product contribution of the paper is also to point out how those two properties can be useful, even outside the clustering applications.

---

> ### Author Response · Authors · 2025-11-20
> **General Reply - Part II**
>
> - **Scalability**:
> Scalability concerns were also raised by Reviewers **AYHy**, **jxWS**, and **Skq7**. While many existing computational frameworks address clustering through mixture estimation, they typically lack strict convergence guarantees to **global** optima. In contrast, our approach recovers globally optimal estimates based on a rigorous distance-optimization criterion. It is true that generic SDPs can face scalability limitations; however, there now exist several specialized SDP solvers with remarkable scalability, and active research efforts (also within the ML community (e.g., Dathathri, 2020)) are deployed to improve their efficiency. Given the large modelization capacity of our framework, developing solvers that exploit the problem structure specific to our formulation represents a natural and promising direction for future work.
> In addition, as acknowledged by Reviewer **Skq7**, we already propose in the Appendix an alternative projection-based strategy using univariate approximations. Thanks to the feedback from Reviewer **jxWS**, we have realized that this idea is related (and can be viewed as an extension) to the well-established sliced Wasserstein distance. Given the space constraints and the promising performance of this approach for accurately estimating the number of clusters on the MNIST dataset (see Table 1), we believe that a dedicated, in-depth analysis of this method constitutes an exciting and original avenue for future research.
>
> -- Sumanth Dathathri et al. Enabling certification of verification-agnostic networks via memory-efficient semidefinite programming. NeurIPS, 2020.
>
> For more detailed responses regarding all specific comments, we refer the Reviewers to the individual response sections.
>
> We greatly appreciate the constructive feedback and will do our best to incorporate these insights to enhance the clarity, completeness, and applicability of our work.

---

### Author Response · Authors · 2025-12-02
**Post-Rebuttal Summary**

Dear **Area Chair** and **Reviewers**,

As the rebuttal period comes to an end, we would like once again to thank you for your
valuable time and constructive feedback, which has helped us improve the quality of our submission.

We would like to take this opportunity to briefly summarize the discussion, highlight the main
points that were raised, and explain how we have addressed them in our responses and revisions.

--------

We recall that our paper introduces a framework for identifying, within a given parametric mixture family, the mixture that is closest to a target measure in the sense of a regularized W2/TV distance.

Namely, we present **two distance-dependent regularized hierarchies tailored to multivariate (not-necessarily Gaussian) mixtures**, and propose an **algorithm** which enjoys **finite convergence guarantees** for jointly estimating (i) **the mixture order** and (ii) **component parameters**. We do so under **minimal assumptions** on the target measure, mixing measure, and parametric family: we do not require the target measure to be compactly supported, the mixing measure to be discrete, or the mixture components to have known or isotropic variances.  As such, our method can be employed either as a **standalone procedure** or as a **pre-processing step** for other ML algorithms, providing high-quality, globally certified (TV/W2-optimal) estimates of the mixture order and parameters. We emphasize that, while clustering is a natural illustrative application of our methodology, our methodology is more general and can be coupled with any ML algorithm that requires high-quality estimates of distances between probability distributions.

Therefore, to the best of our knowledge, no other method provides **globally certified mixture learning** with a comparable level of generality. Taken together with our empirical evaluations, we believe this constitutes a novel and substantive contribution that is **well aligned** with ICLR’s scope.

Given the **scarcity of rigorous global optimization methods within the ML community**, we believe that our work clearly illustrates the importance and potential of such approaches. We also hope it will encourage further **interaction** between the global optimization and ML communities, ultimately leading to highly efficient and scalable tools for global optimization, with applications **well beyond clustering**.

-------
We also invite you to consult our previous replies (**General Reply - Part I, General Reply - Part II**, as well as **individual replies**) as they contain more elaborated answers. In particular,

---> Reviewer **jxWS** found the revised version more accessible and raised their initial score.

--->  Reviewers **8vsn** and **AYHy** mainly expressed concerns about the presentation and suitability. We are strongly convinced that the revised version, as well as our replies, properly address all of these points and that this would be reflected in their assessments were they able to update them.

---> Reviewer **Skq7**’s main concern was the suitability of our work for ICLR. As we have argued in the paper and throughout the discussion period, we believe that our submission **brings important tools from the global optimization community to the attention of the ICLR community**, with **minimal technical overhead** and clear illustrative potential.

---------------------

Taken together, we estimate these clarifications convey that our work offers a **rigorous, novel, and relevant contribution for globally certified mixture
learning** that aligns well with ICLR’s scope and objectives.

We thank you all once again for your time and involvement in improving the clarity and quality of our work.

---

### Meta-Review · Area_Chair_fM66 · 2026-01-07

**Summary:**

The paper applies a recently published approach to the approximation of W1 and TV distances between measures (Laserre 2024) to the specific case of approximating a given density by a finite mixture. The basic idea is to solve a semidefinite program based on a subset of the moments of the finite mixture (Skq7). They tested the proposed methods on a range of unsupervised clustering tasks on the underlying parametric mixture family, which produces mixture estimates that can significantly improve the performance of standard clustering algorithms (8vsn).

This is a theoretical paper about using SDP to solve mixture model better at a time when both mixture models and the value of exact solutions are challenged in the ML community. 3 of 4 reviewers (AYHy, Skq7, jxWS) questioned if ICLR is the venue for this work, where they forecast that most of the attendees will not have the background to use and understand this work. This is an important point since even if accepted and presented at the conference, there is limited value to the audience and to the authors.

The practical value must also be limited since this method naturally will not scale well with dimension (this is not addressed by the authors who pointed to their MNIST experiment that first reduces MNIST to to 2-4 dim using PCA). The AC agrees with the reviewers that this work should reach the appropriate audience and be evaluated by them. For acceptance at ICLR though, at least one reviewer should appreciate the contributions and advocate for the paper. For this paper, the matching system worked reasonably well, where the paper is matched with reviewers who understood the main message. The reviewers choose to evaluate the work on its relevance to the community and practical impact rather than the novelty or rigor of the technique.

**Reviewer Concerns:**

not really addressed or addressable

**Reviewer Scores:**

no change

---

### Decision · Program_Chairs · 2026-01-26

Reject